# A Survey on Big IoT Data Indexing: Potential Solutions, Recent Advancements, and Open Issues

**Zineddine Kouahla** [1], **Ala-Eddine Benrazek** [1], **Mohamed Amine Ferrag** [1,*], **Brahim Farou** [1], **Hamid Seridi** [1], **Muhammet Kurulay** [2], **Adeel Anjum** [3] **and Alia Asheralieva** [3]

[1] Labstic Laboratory, Department of Computer Science, Guelma University, Guelma 24000, Algeria; kouahla.zineddine@univ-guelma.dz (Z.K.); benrazek.alaeddine@univ-guelma.dz (A.-E.B.); farou.brahim@univ-guelma.dz (B.F.); seridi.hamid@univ-guelma.dz (H.S.)
[2] Department of Mathematics Engineering, University of Yildiz Technical, Istanbul 34349, Turkey; mkurulay@yildiz.edu.tr
[3] Department of Computer Science and Engineering, Southern University of Science and Technology, Shenzhen 518055, China; adeelanjum2001@hotmail.com (A.A.); asheralievaa@sustech.edu.cn (A.A.)
* Correspondence: ferrag.mohamedamine@univ-guelma.dz

**Abstract:** The past decade has been characterized by the growing volumes of data due to the widespread use of the Internet of Things (IoT) applications, which introduced many challenges for efficient data storage and management. Thus, the efficient indexing and searching of large data collections is a very topical and urgent issue. Such solutions can provide users with valuable information about IoT data. However, efficient retrieval and management of such information in terms of index size and search time require optimization of indexing schemes which is rather difficult to implement. The purpose of this paper is to examine and review existing indexing techniques for large-scale data. A taxonomy of indexing techniques is proposed to enable researchers to understand and select the techniques that will serve as a basis for designing a new indexing scheme. The real-world applications of the existing indexing techniques in different areas, such as health, business, scientific experiments, and social networks, are presented. Open problems and research challenges, e.g., privacy and large-scale data mining, are also discussed.

**Keywords:** big data; Internet of Things; indexing; information retrieval; query

## 1. Introduction

Widespread utilization of Internet of Things (IoT) systems and applications has resulted in the massive data expansion promising greater benefits for businesses and individuals, but also introducing significant challenges for big data analytics. Such an expansion also plays an important role in the dynamics of large data. In particular, large data can be classified according to their volume, variety and velocity ("3V's" for short). These categories were first introduced by Gartner, Inc. to highlight some elements of the challenges associated with large volume data [1–5]. Afterwards, veracity and value have been incorporated as two additional categories ("5V's") [6,7]. Others have also expanded this big data category to 6V's and 7V's [8]. The capacity to process and use large amounts of IoT data, e.g., smart-city, smart-grid, e-health, Internet of Vehicles (IoV), Internet of Video Things (IoVT), agriculture, etc., is a key factor in the success of a project [9–11]. The process of indexing large amounts of IoT data comprises many phases where a range of IoT data is analyzed to highlight changes.

Throughout the last decade, privacy violations have increased dramatically. Although private data is emerging as an extremely important resource for business development, the activities of collecting, processing and personal trading data are leading to increasing privacy disclosure risks, and numerous privacy disclosure incidents [12,13].

The use of data indexing and IoT in large datasets is extremely resource-intensive, and IoT may be an exceptional alternative. The process of merging technologies increases

the possibilities of deploying IoT in more effective domains that attempt to extend the ideas of social interconnectedness to IoT; consider the most common ones, namely, the Social Internet of Things (SIoT) [14,15], the Multiple IoT Environment (MIE) [15,16], and the Multiple Internets of Things (MIoT) [15,17,18]. Figure 1 represents the process of discovering and searching large IoT indexing data from different IoT devices under a three-tier fog computing architecture. Implementing large-scale IoT data integration solutions in a fog computing architecture [19–21] can help overcome data and indexing issues.

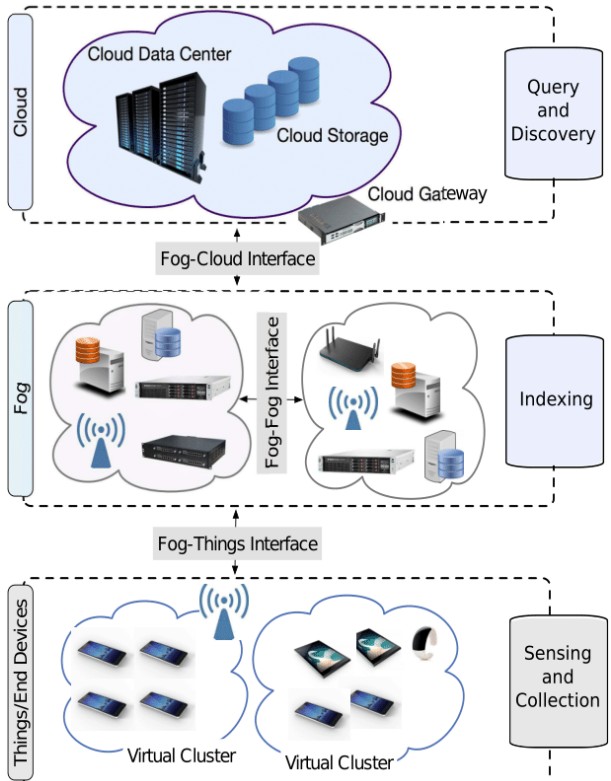

**Figure 1.** Discoverable and searchable of Big IoT Data indexing from different IoT devices under three-tier fog computing architecture.

In addition, it can also contribute to improving collaboration and communication between different objects in a smart city. Hence, the management and indexing of large-scale data has been the subject of several large-scale data reviews. In this study, the authors focus, however, on large IoT data in the context of indexing a massive amount of data.

## 2. Motivation

The methods used to process IoT data must be efficient. However, the increase in data size with the appearance of new types of data (time series, fingerprints, DNA sequences, documents, etc.) has changed the problem.

Due to the diversity of IoT research, solutions developed in one application environment may not be compatible with others. For this reason, various survey papers are presented on IoT and big data for multiple applications see Table 1. A number of investigative papers related to different aspects of data IoT are published to date, covering various definitions of IoT, core technologies, architecture, and different applications IOT, for example [22–25].

**Table 1.** Comparison of past surveys.

| Survey | Year | Architecture | Data Type | Dimension | Complexity | Application | Data Structure | Objectives |
|---|---|---|---|---|---|---|---|---|
| S. Pattar et al. [26] | 2018 | Yes | Yes | Partial | Partial | No | Partial | • Present a review of leading research methods for IoT and classify them according to their design principle and research approaches such as IoT data. |
| Mohammadi et al. [27] | 2018 | Yes | Yes | No | No | Yes | No | • Identify the characteristics of the IoT data<br>• Focus on the challenges of research for a successful fusion of Deep Learning and IoT applications.<br>• Review current methods of advanced DL and their applicability in the field of IoT, both for large datasets and for continuous analysis. |
| Saha et al. [28] | 2018 | No | No | No | No | Yes | No | • Propose a taxonomy of Big Data technologies in IoT fields.<br>• Suggest large-scale data technologies applicable in the field of IoT.<br>• Discuss the advantages and disadvantages of large-scale data technologies in IoT. |
| Shabnam et al. [29] | 2018 | Yes | Yes | No | No | Yes | No | • Combine the systematic mapping and literature review<br>• Propose a taxonomy of three categories : Architecture and platform, framework and application. |
| R. Ettiyan et al. [30] | 2020 | Yes | No | No | No | Yes | Yes | • Examine a various kind of applications including the Healthcare Management System experimented and implemented via IoT in recent years. |
| Eceiza et al. [31] | 2021 | Yes | Partial | Partiel | No | No | Partial | • Present a review of fuzzy techniques and proposals, and their applications to embedded IoT devices. Furthermore, propose future research directions, highlighting the gaps identified in the analysis. |
| Wei et al. [32] | 2021 | Yes | No | No | No | Yes | No | • Presents a comprehensive review of the application of ML techniques for the analysis of important IOT data in the healthcare sector.<br>• Discuss the benefits and challenges of existing techniques. |
| Baofeng et al. [33] | 2021 | Yes | No | No | No | Yes | No | • Examine the benefits, applications of critical infrastructure technologies-NIB-, typical use cases and IdE-based development trends. |
| A.Shah et al. [34] | 2021 | Yes | No | No | No | Yes | No | • Investigate CME and network splitting for the provision of 5G service-orienteduse.<br>• Discusses recent progress in the implementation of E2E network slicing, its core technologies, solutions, and current standardization efforts |
| S.Amin et al. [35] | 2021 | Yes | No | No | No | Yes | No | • Analyze the existing and evolving edge computing architectures and techniques for smart healthcare and recognize the demands and challenges of different application scenarios.<br>• Examine edge intelligence that targets health data classification with the tracking and identification of vital signs using state-of-the-art deep learning techniques.<br>• Presents a comprehensive analysis of the use of cutting-edge artificial intelligence-based classification and prediction techniques employed for edge intelligence. |
| Chegini et al. [36] | 2021 | Yes | No | No | No | Yes | No | • Examine, studie and analyze automatic functions.<br>• Demonstrate the automatic functions through these searches according to each challenge. |
| Our survey | / | Yes | Yes | Yes | Yes | Yes | Yes | • Identify and evaluate the main data indexing techniques in the IoT system.<br>• Classify the indexing techniques used in large data.<br>• Design a taxonomy and analyze the indexing techniques according to the indexing needs of large data.<br>• Provide a structural comparison based on the construction and search algorithms related to these techniques<br>• Explore the opportunities and challenges for each of the reviewed methods and IoT environments.<br>• Review the emerging areas that would intrinsically benefit from Big data indexing and IoT. |

The study of [22] reviews the state of art of different data mining techniques used in large and small-scale IoT applications. It provides the general context and reviews several related applications and technologies. However, it lacks the comparative study between the structures used and their efficiencies.

Other work [27,37] has studied the convergence of data mining with IoT. The study by [38] examined the power of large IoT data analysis in IoT applications. Along with the discussion on data analysis, method and techniques of IoT, they also presented a cloud oriented data architecture.

Most of this research is focused on technology, knowledge extraction or analysis. Some also have applications, but are specific to a particular application. This study presents a systematic and detailed review, oriented towards indexing structures, of various data construction and extraction algorithms that are well-used in an IoT environment. We have focused this work towards the main contributions of our research work are the following:

- Identify and evaluate the main data indexing techniques in the IoT system.
- Classify the indexing techniques used in large data.
- Provide a structural comparison based on the construction and search algorithms related to these techniques.
- Design a taxonomy and analyze the indexing techniques according to the indexing needs of large data.
- Explore the opportunities and challenges for each of the reviewed methods and IoT environments.
- Review the emerging areas that would intrinsically benefit from Big data indexing and IoT.

The amount of information about IoT has multiplied considerably in recent years. Therefore, having an efficient search system is currently one of the main challenges for researchers.

In particular, based on the prior work in [39], an overview of the high requirements of massive IoT data indexing is presented and a new taxonomy of indexing techniques is proposed. In addition, a comprehensive review of existing research on indexing techniques is presented for a better understanding of the differences between the Big IoT data indexing techniques. A thorough comparison of the existing research on indexing techniques is also provided according to the datasets used, types, advantages, disadvantages and challenges. The paper presents a comparative analysis of multidimensional indexing approaches and metric access methods. Finally, an enumeration of existing research challenges and potential opportunities for future research directions in the area of data indexing for large-scale IoT projects is presented.

### 2.1. Methodology for Selecting the Research Papers

The identification of literature for analysis in this paper was based on a keyword search, namely, "Indexing and searching", "Big IoT data indexing", "Indexing technique", and "indexing framework". Searching for these keywords in academic databases such as SCOPUS, Web of Science, and ACM Digital Library, an initial set of relevant sources were located. The search process produced a significant number of results. Although a systematic collection of literature has been performed, recent research has shown that relevant primary sources can be missed during searches and that multiple researchers working on the same methodology may collect differing bodies of articles. However, only proposed indexing and searching techniques for IoT applications were collected. Secondly, each collected source was evaluated against the following criteria: (1) reputation, (2) relevance, (3) originality, (4) date of publication, and (5) most influential papers in the field. The higher the overall score, the higher the source was ranked on our list. Using this ranking system allowed the prioritization of sources. The final pool of papers consists of the most important papers in the field of IoT data that focus on indexing and searching of large data collections as their objective. Our search started on 1 January 2019 and continued until the submission date of this paper.

*2.2. Survey Organization*

This survey article is organized around several sections, as shown in Figure 2. It consists of eight main sections:

- Section 1
    - We present the reasons of the emergence of Big IoT Data and why indexing techniques are required.
    - We illustrate the process of discovering and searching large IoT indexing data from different modern IoT paradigms using a three-tier fog computing architecture.
    - We also provide a summary of existing literature surveys and what are the main gaps compared to our review.
    - We highlight the different contribution of the proposed survey and they are organized in the manuscript

- Section 2
    - We present and describe the indexing requirements.
    - We explain the advantages of metric space and what is actually added to the indexing techniques with regard to the multidimensional space.
    - We highlight the critical importance of similarity queries in IoT applications involving large volumes of data and complex objects.

- Section 3
    - We first present our proposed taxonomy of existing indexing techniques in the literature
    - We provide a detailed description of the majority of the indexing technique under the proposed taxonomy.
    - We provide a comparative performance study of recent indexing techniques and their ability to solve Big IoT Data indexing problems
    - We summarize our analytical and comparative study for each indexing technique type in the Tables 1–15

- Section 4
    - We recall and study in depth several important techniques of multidimensional space
    - We provide the main challenges for each indexing structure and its potential solutions

- Section 5
    - We recall and study in depth several important techniques of metric space
    - We provide the main challenges for each indexing structure and its potential solutions solutions

- Section 6
    - We identify different directions for future research, which we believe are relevant to our work
    - We briefly define each search direction and how indexing techniques can benefit from it

- Section 7
    - We provide a brief summary of our study and we highlight the most important issues that need to be addressed as soon as possible.

- Section 8
    - We recall the main objective of our survey.
    - We provide a quick overview of the work provided by our manuscript

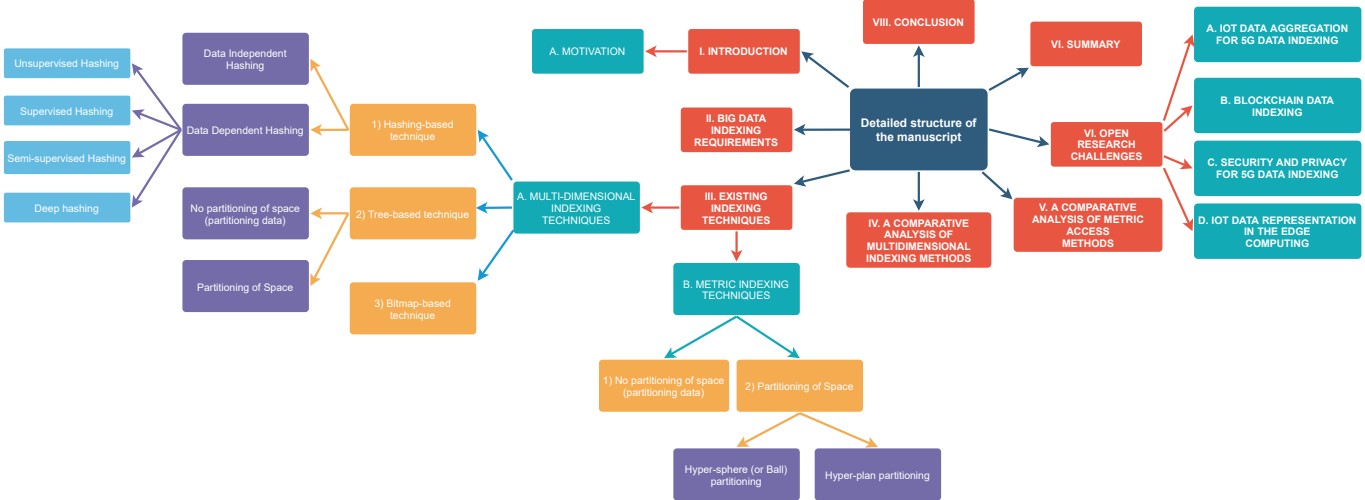

**BIG DATA INDEXING REQUIREMENTS**
**Section II**

**(1)** We present and describe the indexing requirements.

**(2)** We explain the advantages of metric space and what is actually added to the indexing techniques with regard to the multidimensional space.

**(3)** We highlight the critical importance of similarity queries in IoT applications involving large volumes of data and complex objects.

**A COMPARATIVE ANALYSIS OF MULTIDIMENSIONAL INDEXING METHODS**
**Section IV**

**(1)** We recall and study in depth several important techniques of multi-dimensional space

**(2)** We provide the main challenges for each indexing structure and its potential solutions

**OPEN RESEARCH CHALLENGES**
**Section VI**

**(1)** We identify different directions for future research, which we believe are relevant to our work

**(2)** We briefly define each search direction and how indexing techniques can benefit from it

**CONCLUSION**
**Section VIII**

**(1)** We recall the main objective of our survey
**(2)** We provide a quick overview of the work provided by our manuscript

**INTRODUCTION**
**Section I**

**(1)** We present the reasons of the emergence of Big IoT Data and why indexing techniques are required.

**(2)** We illustrate the process of discovering and searching large IoT indexing data from different modern IoT paradigms using a three-tier fog computing architecture.

**(3)** We also provide a summary of existing literature surveys and what are the main gaps compared to our review.

**(4)** We highlihgt the defferent contribution of the proposed survey and they are organized in the manuscipt

**EXISTING INDEXING TECHNIQUES**
**Section III**

**(1)** We first present our proposed taxonomy of existing indexing techniques in the literature

**(2)** We provide a detailed description of the majority of the indexing technique under the proposed taxonomy.

**(3)** We provide a comparative performance study of recent indexing techniques and their ability to solve Big IoT Data indexing problems

**(4)** We summarize our analytical and comparative study for each indexing technique type in the tables (1-15)

**A COMPARATIVE ANALYSIS OF METRIC ACCESS METHODS**
**Section V**

**(1)** We recall and study in depth several important techniques of metric space

**(2)** We provide the main challenges for each indexing structure and its potential solutions

**SUMMARY**
**Section VII**

**(1)** We provide a brief summary of our study and we highlight the most important issues that need to be addressed as soon as possible.

**Figure 2.** Organization of the survey.

To help readers navigate this paper, Figure 3 provides a detailed structure of the survey.

**Figure 3.** Detailed structure of the survey.

## 3. Big IoT Data

Big Data results from the significant growth and data accumulation of Internet operations and online applications such as social networks and video streaming [40,41]. However, in the context of IoT, different end-devices such as Personal Computers (PCs), smartphones, Global Positioning System (GPS) devices, sensors, and Radio Frequency Identification (RFID) devices, monitoring devices, etc. used in different applications such as healthcare, manufacturing, industry, smart homes, smart cities, etc. collect a large amount of data continuously, making IoT one of the main sources of Big data. Besides, it is also important to mention that crowd-sourcing and crowd-sensing mechanisms and tools play a more important role in Big IoT Data collection nowadays [42,43].

The merging of Big data and IoT data (or Big IoT data) created new features in addition to the Big data features discussed in [44–48]. These additional data characteristics provoke the implementation of new data management techniques that consider the characteristics of Big IoT Data, presented in Figure 4. In other words, efficient Big IoT Data indexing and searching in large data collections is, therefore, a critical issue that requires choosing the appropriate structure.

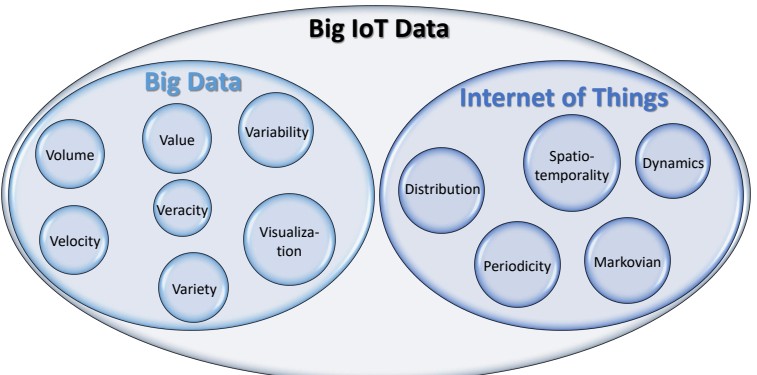

**Figure 4.** Big IoT data's characteristics.

## 4. Big Data Indexing Requirements

Some recent studies on indexing techniques have highlighted how to optimize the search performance in large datasets with greater efficiency. This section discusses and describes some of the indexing needs that are more challenging than traditional data. Then, each indexing technique is analyzed based on these constraints to determine its applicability on a large scale.

The challenges of designing the indexing techniques are related to the need for generalization of multidimensional spaces to metric spaces and the additional constraints on the set of feasible solutions. The constraints in Figure 5 may include the constraints on data independence, scalability, or efficiency [49].

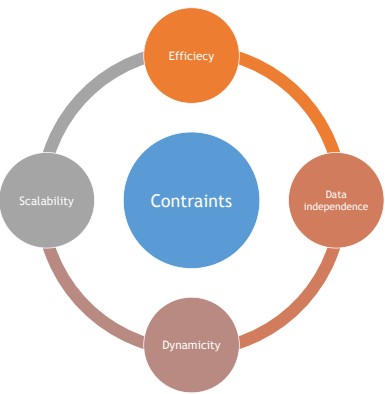

**Figure 5.** Big Data indexing requirements.

In general, the objects which require indexing are more complex than mere vectors [50–54]. This shifts the focus of indexing from multidimensional spaces to metric spaces. Formally, a metric space is defined for quantifying similarities or different elements through a given distance in such a way that smaller distances may correspond to more similar elements. Metric spaces are therefore a very general concept and can be applied to vectors as well as objects such as strings and graphs that can not easily be represented as vectors [55]. Several similarity measures exist for various types of objects, such as Minkowski distances. Manhattan and Euclidean distances are best known and can be used for any form of vector data, such as color histograms in multimedia databases.

Similarity queries are a very important operation in IoT applications involving large data volumes and complex objects. They focus on finding objects in a dataset similar to a query object, based on a similarity measure. In metric space, the similarity query refers to the selection of objects in a dataset $\mathcal{O}$ that are at a certain distance $d(\cdot, \cdot)$ from a given point $o_q$.

The main factor that affects the efficiency of the indexing algorithm when the dimension is increased is the dimensionality-curse problem [56,57]. The approaches available in the literature have proven to be unreliable, making it difficult to index, manage and analyze large volumes of data. This is due to the inherent deficiencies of spatial partitioning and also to the factor of overlap between regions. This question therefore remains open for future research.

## 5. Existing Indexing Techniques

Section 2 outlines requirements based on the need to index and retrieve increasingly numerous and complex data [58]. Thus, small data collections that include simple objects can be easily processed. However, managing large databases, like most databases used today, requires more sophisticated techniques, especially when they contain complex data types. The objects to be indexed are sometimes more complex than simple vectors (homogeneous—e.g., vector spaces; or heterogeneous—e.g., tuplets in a relational database) [59–62].

Several indexing techniques have been introduced to address the problems of indexing large data. This paper provides a comparative performance study of recent indexing techniques and their ability to solve large data indexing problems. In addition, these techniques are examined according to a proposed taxonomy (see Figure 6).

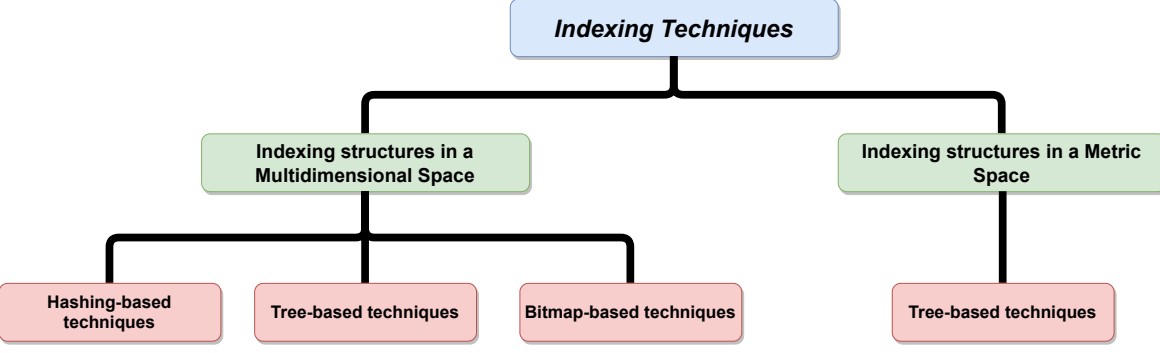

**Figure 6.** Global taxonomy of indexing techniques.

Figure 6 describes the classification of indexing techniques according to space. Indexing techniques can be classified into two main categories: (i) multidimensional space and (ii) metric space.

### 5.1. Multidimensional Indexing Techniques

A multidimensional space is defined when the elements of the set are considered as vectors (i.e., the data has a given number of dimensions), homogeneous or heterogeneous, whose components are totally ordered. Thus, indexing techniques in multidimensional spaces can be classified into three main types according to the type of structure used:

(1) hashing-based technique, (2) tree-based technique, and (3) bitmap-based technique. In the following, we review the three types of multidimensional indexing techniques.

### 5.1.1. Hashing-Based Technique

This is a more popular technique in the field of multidimensional data indexing due to its ability to transform a data item into a low-dimensional representation (short code composed of a few bits) [63]. Hashing-based indexing structures are more efficient in terms of time and storage space [64] and can detect duplicate data in a large dataset [65]. There are many methods based on the hashing technique applied to several real applications, such as computer vision, information retrieval and analysis (e.g., images, videos, documents) [66]. According to Figure 7, hash-based indexing structures can be classified into two main streams: *data independent hashing* and *data dependent hashing (or learning-based hashing)*.

**Data Independent Hashing:** Among the data independent hashing methods, the Locality-Sensitive Hashing (LSH) developed by Gionis et al. [67] is the most popular in the literature. It allows retrieving the sufficient set of Approximate Nearest Neighbors (ANNs) in high dimensional space. One of the main criteria of the LSH techniques family is the hash function which returns with high probabilities, the same bit for close data points in the original space [68]. Since LSH's proposal, several variants have been proposed to improve the SLH method as: MultiProbe LSH [69,70], BayesLSH [71], Boosted LSH [72], Super-bit LSH [73], Non-metric LSH [74], Kernelized LSH (KLSH) [75] and Asymmetric LSH (ALSH) [76]. However, LSH-based techniques suffer from problems of increasing storage costs and search time due to the long binary codes and high hash functions required when the recovery precision is improved [68]. In general, data-independent hash methods are well suited for small data, but they are not sufficient to handle large data. Table 2 shows a summary of the advantages and disadvantages of the above methods, as well as their challenges.

**Data Dependent Hashing:** In the hash stream depending on the data, several methods have been proposed to overcome the problems and limitations of data independent hash methods. These methods are classified into three categories according to the degree of supervision, namely: (i) unsupervised hashing, (ii) supervised hashing, (iii) semi-supervised hashing, and (iiii) deep hashing.

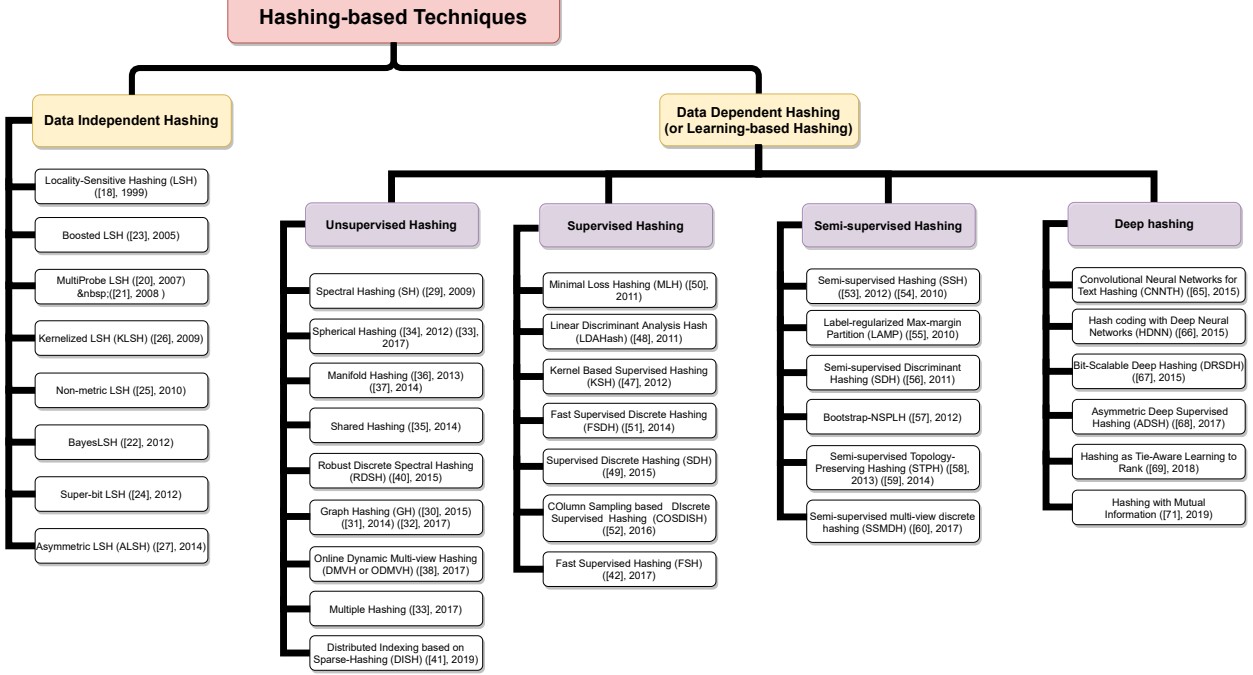

**Figure 7.** Taxonomy of hashing-based indexing techniques.

**Table 2.** Summary of advantage and disadvantage of data independent hashing techniques.

| Proposition | Refs | Advantages | Disadvantages and Challenges | |
|---|---|---|---|---|
| LHS | [67] | • Returns with high probabilities the same bit for nearby data points in the original space by storing similar data in the same bucket | • High storage cost<br>• High search time<br>• Not sufficient to processes high dimensional data. | |
| MultiProbe LSH | [69,70] | • Reduce the number of hash table, therefore, reduce space and time compared to LSH method | • Insufficient number of neighborhood candidates to respond to KNN's requests | |
| Kernelized LSH | [75] | • Search for approximate similarity in sublinear time<br>• No data distribution or data entry assumptions are required | • High memory consumption<br><br>• The search for the nearest neighbor is very difficult for high dimensional data | Unsuitable to process large data |
| BayesLSH | [71] | • High quality of search results | • Less effective performance | |
| Super-bit LSH | [73] | • Significant error reduction<br>• More effective for approximate nearest neighbor recovery | • Requires long hash codes and more hash tables<br>• High cost of space and time | |
| Asymmetric LSH | [76] | • Simple and easy<br>• Efficient for maximum inner product research | • Does not support exact search | |

**Unsupervised Hashing:** For higher precision in the design of compact hash codes, unsupervised hashing methods aim to integrate data properties such as distributions and multiple structures [77]. Reference methods include spectral hashing [78], graph hashing [79–81], multiple hashing [82], spherical hashing [82,83], shared hashing [84], manifold hashing [85,86], etc.

Recently, a novel unsupervised online hashing method for online image retrieval was proposed by Liang et al. [87], called Online Dynamic Multi-View Hashing (ODMVH or DMVH) capable of adaptively increasing hash codes according to dynamic changes in the image. These hashing techniques also use multi-view features to achieve more efficient hashing performance. DMVH has limited performance because it is an unsupervised method and has not exploited any discriminative semantic information [88].

Yang et al. [89] developed a novel unsupervised hashing approach, named Robust Discrete Spectral Hashing (RDSH) to facilitate large-scale semantic indexing of image data. RDSH can simultaneously learn discrete binary codes and robust hash functions in a unified model. Due to the difficulty of the latter, the authors included the offline process for the learning binary codes as well as the coding functions and the online procedure for indexing images with semantic annotations. Initially, the real value representation is learned from the original space of the entities using methods such as Spectral Hashing (SH). Then, the real representation is transformed into binary codes through binarization based on learning. Several experiments have been performed on various real-world image datasets to demonstrate its effectiveness in large-scale semantic indexing approaches. Compared to locality-sensitive hashing, the spectral hashing generates a very compact hash code, but it is not appropriate for a large and dynamic database. A new Distributed Indexing method based on Sparse-Hashing (DISH) in cloud computing was developed by André et al. [90] to address the difficulties associated with distributing an index of high-dimensional feature vectors to multiple index nodes and search for large-scale distributed images. DISH allows documents and queries to be distributed in a balanced and redundant way between nodes. Table 3 provides a comparison of the technique discussed above.

**Supervised hashing:** Supervised hashing methods are based on machine learning techniques such as decision tree [91] and neural networks [92]. These methods aim to generate intelligent indexes that can predict the unknown behavior of the data [93,94]. The supervised hashing methods allow to treat semantic similarities as well as the search for medical images on a large scale [77,95]. Many representative methods have used some form of supervision to design more efficient the hash functions: Kernel Based Supervised Hashing (KSH) [96], Linear Discriminant Analysis Hash (LDAHash) [97], Supervised Discrete Hashing (SDH) [98], Minimal Loss Hashing (MLH) [99], Fast Supervised Hashing (FSH) [91] and Fast Supervised Discrete Hashing (FSDH) [100].

**Table 3.** Summary of advantage and disadvantage of unsupervised hashing techniques.

| Proposition | Refs | Advantages | Disadvantages | |
|---|---|---|---|---|
| Spectral Hashing | [78] | • Does not require any labeled data<br>• Solve a difficult non-linear optimization problem with a global optimum | • The assumption of a uniform distribution of data is usually not applicable in most cases of real-world data<br>• Cannot directly applied in the kernel space<br>• Does not work very well for high-dimensional data | • Less efficient than a (semi-) supervised hashing technique |
| Spherical Hashing | [82,83] | • Ensuring high accuracy and a highly scalable search for the nearest neighbor | • Not sufficient for high-dimensional data<br>• Limited performance.<br>• Requires an expensive learning process to learn the hash functions | • Unsuitable to process large data |
| Robust Discrete Spectral Hashing | [89] | • Robust hash functions<br>• Very compact hash code compared to LSH | • Not appropriate for a large and dynamic database | |
| Graph Hashing | [79,80] | • Suitable for large-scale applications<br>• high search precision | • Inefficient in the search of nearest neighbors<br>• High learning costs | |
| Online Dynamic Multi-view Hashing | [87] | • More efficient hashing performance | • Limited performance | |
| Distributed Indexing based on Sparse-Hashing | [90] | • Distribution of requests in a balanced way | • High cost time | |

Liu et al. [96] proposed a supervised hash method with kernels (KSH) in the Hamming space, where the hash codes obtained for similar data are similar hash codes (minimizes similar pairs) and for different data, the hash codes obtained are different hash codes (maximizes dissimilar pairs). Kang et al. [101] proposed a discrete supervised hashing method, called Column Sampling based on Discrete Supervised Hashing (COSDISH). COSDISH operates in an iterative way, and in each iteration, several columns are first sampled from the semantic similarity matrix and then the hashing code is decomposed into two parts and alternately optimize them in a discrete way. Compared to FSH [91], which cannot use all training points due to time complexity, COSDISH is capable to use all training data points. Table 4 compares several supervised hashing techniques.

**Table 4.** Summary of advantage and disadvantage of supervised hashing techniques.

| Proposition | Refs | Advantages | Disadvantages | |
|---|---|---|---|---|
| Minimal Loss Hashing | [99] | • Efficient and adapts well to long code lengths<br>• Higher search precision | • Training speed very slow<br>Difficult to optimize | • Difficulty of finding the labeling of all data in the database |
| Linear Discriminant Analysis Hash | [97] | • Effective compact hashing<br>• Less memory consumption and calculation cost | • Slower because of the extraction of SIFT descriptors | • Much slower in terms of time and effort compared to unsupervised techniques |
| Kernel Based Supervised Hashing | [96] | • Efficient hash functions<br>• Higher retrieval accuracy | • Not sufficient for high-dimensional descriptors | |
| Fast Supervised Hashing | [91] | • Suboptimal<br>• Fast ANN search | Not use all training points due to the complexity<br>Unsatisfactory performance in real-world applications | • Unsatisfactory performance |
| Fast Supervised Discrete Hashing | [100] | • Highly efficient<br>• Very fast and high precision<br>• Low storage cost | • Require a significant degree of effort in large-scale applications | |
| Supervised Discrete Hashing | [98] | • Effective binary code learning | • Expensive training time<br>• Insufficient precision rate | • Insufficient for high-dimensional data |
| Column sampling based discrete supervised hashing | [101] | • Capable to use all training data points | • Inefficient binary codes | |

**Semi-Supervised hashing:** Due to complexities of the exhaustive search of data labels in the database, semi-supervised hashing methods can use hash functions capable of training on two types of data, whether labeled or unlabeled data (partially labeled). In other words, semi-supervised hashing is a combination of unsupervised and supervised hashing [95]. The goal of the semi-supervised hashing method is to minimize the

empirical error of labelled datasets and improve the binary encoding performance. Semi-supervised hashing methods are able to handle semantic similarity and dissimilarity between data [102] based on non-weighted distance and simple linear mapping. Representative methods include the Semi-Supervised Hashing (SSH) [102,103], which is considered one of the most popular methods, along with Label-regularized Max-margin Partition (LAMP) [104], Semi-supervised Discriminant Hashing (SDH) [105], Bootstrap Sequential Projection Learning for Semi-supervised Nonlinear Hashing (Bootstrap-NSPLH) [106] and Semi-supervised Topology-Preserving Hashing (STPH) [107,108]. Lately, Zhang and Zheng in [109] presented a new semi-supervised hashing named Semi-Supervised Multi-view Discrete Hashing (SSMDH). SSMDH minimizes the loss jointly when using relaxation on learning hashing codes on multi-view data. SSMDH reduces the loss of regression on a portion of the labeled samples, which increases the discrimination ability of the learned hash codes. Table 5 compares several semi-supervised hashing techniques.

**Table 5.** Summary of advantage and disadvantage of semi-supervised hashing techniques.

| Proposition | Refs | Advantages | Disadvantages | |
|---|---|---|---|---|
| Semi-supervised Hashing | [102,103] | • Empirical Error Minimization<br>• Variance and independence of binary codes maximized | • Not suitable for high dimensional data | • Much slower in terms of time and effort compared to unsupervised techniques |
| Label-regularized Max-margin Partition | [104] | • High-quality hash functions | | |
| Semi-supervised Discriminant Hashing | [105] | • Good separation between data labeled in different classes | | |
| Bootstrap-NSPLH | [106] | • Balanced partitioning of data points<br>• Higher performance | • Expensive training time<br>• Require storage space and a large amount of computation | • Impractical for high-dimensional data |
| Semi-supervised multi-view discrete hashing | [109] | • Minimizes the loss jointly on multi-view features when using relaxation on learning hashing codes<br>• Increases the discrimination ability of the learned hash codes | • Not suitable for high dimensional data | |

**Deep Hashing Methods:** Several studies have used deep learning techniques such as in the image classification [110,111] and object detection [112,113] methods. In addition, some hashing methods available in the literature have focused on the adaptation of deep learning techniques and in particular Deep Artificial neural Networks (DANS) to take advantage of deep learning, such as Convolutional Neural Networks for Text Hashing (CNNTH) [114], Simultaneous Feature Learning and Hash Coding with Deep Neural Networks [115], Bit-Scalable Deep Hashing With Regularized Similarity Learning for Image Retrieval and Person Re-Identification (DRSDH) [116], Asymmetric Deep Supervised Hashing (ADSH) [117] and hashing as tie-aware learning to rank [118]. Due to the automatic learning ability of the deep learning methods, deep hashing methods have shown better performance than traditional hashing methods [119]. Hash methods that adapt to in-depth learning can be based on unsupervised or supervised learning, but most of these methods are supervised, with supervised information given with triplet labels [120]. Jiang and Li [117] proposed Asymmetric Deep Supervised Hashing (ADSH) for large-scale nearest neighbor search. ADSH learns a deep hash function only for query points, while the hash codes for database points are directly learned to reduce the training time complexity. Table 6 below shows a comparison of some key advantages and drawbacks of deep hashing techniques.

**Table 6.** Summary of advantage and disadvantage of deep hashing techniques.

| Proposition | Refs | Advantages | Disadvantages | |
|---|---|---|---|---|
| Convolutional Neural Networks for Text Hashing | [114] | • Better performance than traditional hashing methods | • Unsuitable for all real-world domain databases<br>• Not sufficient to processes high dimensional data | • Performance decreases as the dimensionality of the data increases |
| Hash coding with Deep Neural Net | [115] | • Better performance<br>• Good search precision rate | • Demand pairwise similarity labels<br>• Need a more complex configuration | |
| Bit-Scalable Deep Hashing | [116] | • Better performance than traditional hashing methods | • Required labeled data and considerable human efforts | |
| Asymmetric Deep Supervised Hashing | [117] | • Reduce the complexity of training time<br>• High search precision rate | • Learns the hash function only for query points<br>• Higher complexity | |

Between unsupervised hashing and (semi-)supervised hashing, the most significant difference is the availability of label information for learning hash functions [121]. Compared to unsupervised hash methods, supervised methods are much slower in terms of time and effort due to the overload of the training process and due to the absence of label information. Thus, the unsupervised methods have a potential value for practical applications as they do not require any labeled information [122,123]. On the other hand, supervised techniques take into account the advantage of explicit semantic labels of the data, which provides a higher efficiency than unsupervised hashing techniques [64].

In general, to achieve satisfactory performance with data independent methods, many hash tables or long hash codes are required, which often makes them less effective in practice than data dependent methods. For data dependent, hashing methods (unsupervised, supervised and semi-supervised hashing) are needed to new solutions to address the problem of optimization to learn hash functions and hash codes.

5.1.2. Tree-Based Technique

Multidimensional data has a number of dimensions. In metric spaces, this notion disappears, not only does it disappear because the object is only considered as a whole and not as a set of components, but also because some objects are naturally without any perceptible dimension. This is the case of a sequence of characters, a set of elements of any description, a graph, etc.

This section presents some tree indexing techniques. Reference books or syntheses on the subject have been proposed by several other authors. In addition, some authors have considered multidimensional indexing techniques as unsupervised classification methods. It is important to note that in a classification, the classes are not of the same cardinal and that, in a hierarchical classification, not all leaf classes are located at the same depth. Indexing techniques can be classified according to two main approaches:

**No Partitioning of Space (Partitioning Data):** The primary idea of data partitioning consists of creating data packets or clusters, also called "inclusion forms". In the literature, there three data-partitioning methods: those whose Minimum Bounding Regions (MBR) are hyper-cubes, those whose MBRs are hyper-spheres and those whose MBRs are hyper-plane [124]. The main representative techniques of this approach include the B-tree [125] (and its and its variants: B$^+$-tree [126], B*-tree, T-tree [127], UB-tree [128], BUB-tree [129], etc.), R-tree [130], the X-tree [131] and the SR-tree [132].

R-tree is a hierarchical data structures based on B$^+$-tree, where it is used to index spatio-temporal data of n-dimensions. R-tree generates several small minimum bounding rectangles (MBR) [130] to reduce dead spaces. R-tree is a balanced [133] and dynamic structure [130] that is very efficient for range requests [134]. The disadvantages of the R-tree structure reside in the increase in space, time and complexity of the calculations because of overlapping multiple MBR regions [135]. Because of the overlapping, R-tree is inefficient for point location queries which can lead to a degradation of the performance of the search process [136]. Several extensions have been proposed based on the R-tree structure to

address the weaknesses of this structure mentioned above. Among these extensions, it can be noted: R$^+$-tree [137], R$^*$-tree [138], Hilbert R-tree [139] and SS-tree [140].

X-tree (eXtended node-tree) [131] is an R-tree based structure developed to prevent overlap between MBRs through the new proposed node type. These nodes are extended nodes of variable size called Super-nodes (eXtended node). Due to this type of node, X-tree supports the indexing of large data with less overlap and less performance reduction compared to the R-tree structure. X-tree is a hybrid index that consists of a hierarchical part (tree) and a linear part (list). X-tree is a variable structure where size and complexity is difficult to calculate because of their sensitivity to size, distribution of data [141]. In addition, X-tree consumes a lot of memory space for storage and its performance is limited with the data dimension.

R-tree nodes and its variants reduce the number of partitions that occur in the construction of the R-tree and increase the spatial utilization of the R-tree to solve the problem of overlapping, which influences the construction performance and requests efficiency. Yang et al. [142] proposed a new lazy splitting strategy to optimize the R-tree generation process. Bloom Filter Matrix (BFM) is a multidimensional data indexing structure developed by Wang et al. [143] to solve the problem of decreasing index performance for high-dimension data. BFM uses a multidimensional matrix based on the Cartesian product of bloom filters where each filter represents an attribute of the original data. Although the BFM structure demonstrates a multi-attribute data indexing speed and search accuracy, it suffers from a very high space consumption [144], which makes it inadequate for IoT applications where data is massive. For an efficient R-tree index, Wang et al. [145] proposed a new retrieval method, called the Dynamic Clustering Center (DCC) method, which allows choosing the optimal cluster center according to the distance indicator R during the construction of the R-tree spatial index. This technique aims to make R-tree structure more compact, reduce multipath searches and improve search efficiency.

The requirement to classify data flow records such as web traffic flow monitoring, spam detection and intrusion detection is addressed in [146]. A new E-tree indexing structure with a time complexity less than $O(log_n)$ was proposed by Zhang et al. to organize all base classifiers in an ensemble for fast prediction. E-tree used a balanced height structure like an R-tree to reduce the expected prediction time from linear complexity to sublinear complexity. On the other hand, E-tree is automatically updated by the repeated aggregation of new classifiers and the elimination of those that are relevant or obsolete. It therefore adapts well to discover new trends and patterns and undifferentiated data flows [146,147]. E-tree requires a high storage space and maintenance [148] despite the results of the analysis showing the effectiveness of this approach. ER$^+$-tree is a new multidimensional data indexing.

The structure on the cloud-computing infrastructure was proposed by Balasubramanian in [149]. This structure is a hybrid tree structure that combines the benefits of E-tree [146] and R$^+$-tree [137]. The main objective of this structure is to reduce computation time and improve the quality of the similarity search in a cloud-computing environment [149]. The idea of combining these two structures is to create a more efficient structure in terms of balancing and similarity research. The E-tree structure is used to partition the data flow to reduce overload, while the R$^+$-tree structure is used to reduce search time and improve the quality of the similarity search through its Minimum Boundary Rectangle (MBR).

In [150], Jin and Song introduced a tree indexing structure based on R*Q-tree. This approach improves query performance and reduces indexing costs. It is based on the *k*-means clustering algorithm to reorganize nodes between neighboring nodes in the tree. In addition, a new indexing method (SUSHI) was proposed by Günnemann et al. [151]. This method is based on subspace clustering for indexing high dimensional objects, where the construction of the index tree is done in a recursive way. The nodes of each level represent the groups resulting from the subspace clustering method. Wang et al. [152] presented a new method based on searching for the nearest neighborhood to accelerate the

matching of corresponding faces for large-scale facial recognition systems. This method uses the k-means algorithm for clustering data and the Kd-tree structure for cluster storage. However, this technique presents, in addition to all the advantages, a problem linked to the complexity of the closing forms, which leads to an increase in the costs of insertion and search operations. Table 7 analyzes multidimensional indexing techniques based on data partitioning, taking into account dataset type, data dimension, indexing nature, and complexity as comparison metrics and Table 8 shows the advantages and disadvantages of these techniques, as well as their challenges.

**Table 7.** Analysis of multidimensional indexing techniques based on data partitioning.

| Proposition | Refs | Dataset Type | Data Dimension | Indexing Nature | Complexity (BigO) | |
| --- | --- | --- | --- | --- | --- | --- |
| | | | | | Insertion/Deletion | Search |
| B-tree | [125] | Temporal | One-dimensional | | $O(n \log(n))$ | $O(n \log(n))$ |
| B+-tree | [126] | | | | $O(n \log(n)))$ | $O(n \log(n))$ |
| B*-tree | [127] | | | | $O(n \log(n))$ | $O(n \log(n))$ |
| T-tree | [127] | | | | $O(2n \log(n))$ | $O(n \log(n))$ |
| UB-tree | [128] | Spatio temporal data | Multi-dimensional | Dynamic | $O(n \log(n))$ | $O(n \log(n)))$ |
| PaIndex | [153] | | | | $O(n \log(n))$ | $O(n \log(n))$ |
| MLB+-tree | [154] | Seismic data | | | $O(n \log(n))$ | $O(n \log(n))$ |
| SR-tree | [132] | Image feature vectors | | | $O(n \log_3(n))$ | $O(n \log_3(n))$ |
| E-tree | [146] | Spatial | | | $<O(n \log(n))$ | Not estimated |
| ER+-tree | [149] | OpinRank Review | | | Not estimated | Not estimated |
| SUSHI | [151] | Color histogram and Synthetic data | | | $O(n^2 \log(n))$ | Not estimated |
| R-tree | [130] | Geographical and Multi-media | | | $O(dn \log(n))$ | $O(n \log(n))$ |
| R+-tree | [137] | | | | $O(n \log(n))$ | $O(n \log(n))$ |
| R*-tree | [138] | | | | $O(n \log(n))+$ *Re-insertion complexity* | $O(n \log(n))$ |
| Hilbert R-tree | [139] | Spatial | | | $O(\log(n) + M \log(n))$ | $O(n \log(n))$ |
| SS-tree | [140] | Multi-media data | | | $O(n \log(n))$ + *Re-insertion complexity* | $O(n \log(n))$ |
| BFM & R-tree | [143] | Not mentioned | | | Not estimated | Not estimated |
| DCC & R-tree | [145] | Medical data | | | $O(nkt)$ | $O(n \log(n))$ |
| X-tree | [131] | Spatial data and Synthetic data | | | Not estimated | Not estimated |
| aX-tree | [155] | Spatial data | | | Not estimated | Not estimated |
| X+-tree | [156] | Spatial data | | | Not estimated | Not estimated |
| R*Q-tree | [150] | Special data | | | $O((kndt)(n \log(n)))$ | Not estimated |
| BB-tree | [157] | Synthetic data, Sensor data and Genomic | | | Not estimated | $O(h \log(k) + b_{max}m)$ for exact-match queries |

**Partitioning of Space:** In this category, indexing techniques are based on space partitioning into sub-spaces (or cells) where each sub-space contains a subset of data. Unlike indexing techniques based on data partitioning, this type of partitioning eliminates region intersections. Many existing approaches are proposed in the literature. Reference techniques include for example: Kd-tree [158], Quadtree [159,160], Pyramid [161] and VA-file [162].

**Table 8.** Summary of advantage and disadvantage of multidimensional indexing techniques based on data partitioning.

| Proposition | Refs | Advantages | Disadvantages | |
|---|---|---|---|---|
| B-tree | [125] | • Simple structure<br>• Balanced in insertion and deletion<br>• Efficient for k-nn and range search | • Consumes a lot of computing resources<br>• Requires large storage space<br>• Costly maintenance | • Support only one-dimensional data |
| B+-tree | [126] | • Storage at leaf nodes<br>• Storage cost reduced compared to B-tree | • High complexity<br>• Wasted storage space<br>• Non-optimal node splitting | • Requires a considerable amount of computing resources |
| B*-tree | [127] | • Reduction of node splitting<br>• Less storage space compared to B-tree and B+-tree | • High complexity | • Limited performance |
| T-tree | [127] | • Balanced structure<br>• More efficient memory management, search and update performance than B+-tree | • Requires a considerable amount of space<br>• Inefficient search<br>• The problem of balance is still unresolved | • Degradation on large scale |
| UB-tree | [128] | • Efficient processing of multidimensional requests | • Unsatisfactory for queries covering dead spaces | |
| PaIndex | [153] | • Effective and efficient update and query performance<br>• Structure supports parallel insertions and queries | • Not suitable for large data | • Degradation on large scale |
| MLB+-tree | [154] | • Higher performance on multi-dimensional range queries | • High complexity<br>• Sub-optimal partitioning<br>• Irregular and unpredictable structure | |
| SR-tree | [132] | • Simple construction<br>• Refinement : (intersection S ^R)<br>• Reduced overlap rate | • Complexity of shapes<br>• Costly insertion and search algorithm | |
| E-tree | [146] | • Reduce time from linear to sublinear complexity | • High storage space | |
| ER+-tree | [149] | • Reduce computation time<br>• High quality of search results<br>• More efficient structure | • Costly maintenance<br>• K-nn research is not evaluated<br>• Degradation on large scale | |
| R-tree | [130] | • Creation of filter cells REM<br>• MBR allows you to refine your search<br>• Balanced hierarchical breakdown<br>• Constraint of minimum coverage | • Overlap of REMs<br>• Not effective for point queries<br>• Require high space and time as well as computational complexities | • Degradation of the performance on large scale |
| R+-tree | [137] | • Reduced overlap rate | • Redundancy of objects in nodes<br>• Clipping technique not optimized<br>• More complex construction and maintenance | |
| R*-tree | [138] | • More efficient variant than the R-tree<br>• Reduced overlap rate<br>• Efficient use of space | • Complexity of the re-insertion algorithm and the split of nodes | |
| Hilbert R-tree | [139] | • Good performance results for both searches and updates | Performance deteriorates for larger data | |
| SS-tree | [140] | • Outperforming the R-tree<br>• Calculate the nearest and approximately nearest neighbors efficiently | • High overlap in high-dimension space | |
| BFM & R-tree | [143] | • Solve the problem of decreasing index performance for high-dimensional data | • High space consumption | |
| DCC & R-tree | [145] | • Enhance R-tree's search efficiency<br>• Reduce multipath searches | • Require high space and computational complexities | |
| X-tree | [131] | • Overlap control (overlap-free)<br>• No degeneration of the index<br>• Reduced overlap rate | • Complexity of the max limit<br>• Consumes a lot of memory space<br>• Performance is limited with the data dimension | • Cannot function properly in higher-dimensional data |
| aX-tree | [155] | • Reduce the amount of empty space<br>• Reduced overlap rate<br>• Fast loading and better partitioning of space | • Supports only static data<br>• Require more calculation | |
| X+-tree | [156] | • Reduces the complexity of linear scanning of super nodes compared to X-tree | • Suffers from data redundancy and replication problems | |
| R*Q-tree | [150] | • Improve space utilization<br>• Reduce node overlap and the number of splits | • High complexity<br>• Not suitable for the situation of frequent updates | |
| BB-tree | [157] | • Quasi-balanced structure<br>• Better performance compared to R*-tree, Kd-tree, PH-tree, and VA-file | • Not support the k-nn search | |

Kd-tree (K-dimensional tree) is a binary tree structure for indexing multidimensional data based on partitioning space to k dimension using hyper-planes [158]. The main disadvantage of the Kd-tree is that it is unbalanced because the hyper-plane of space division do not divide the planes in a better position. The latter creates overlaps between neighboring regions, which increases the cost of I/O operations [163,164]. The performance of the Kd-tree structure to meet range requests or Knn requests is limited by data dimensions, where, as the size of the data increases, most tree data is traversed [165,166]. Several extensions have been proposed to address the challenges of the Kd-tree structure, the best known are: Adaptive Kd-tree [167], KdB-tree [168] and SKd-tree [169]. Similar to the Kd-tree, the Quad-tree [159] is the simplest multidimensional index structure, which is mainly used to partition a two-dimensional space by recursively dividing it into quadrants, and it includes several parts index space (each node has four leaf nodes). The Quad-tree is also not balanced because it does not choose the best division of space (horizontal or vertical) as Kd-tree. In addition, Quad-tree does not take into account the spatial distribution of data during the space partitioning phase [170].

Pyramid tree is also a multidimensional data indexing structure. Pyramid tree is based on the partitioning of the data space into 2D pyramids, each of them is cut in a parallel slice at the base of the pyramid forming the data ranges [161,171–173]. Pyramid tree suffers from the degradation of its performance with the increase in the size of the data because the number of pyramids is insufficient to discriminate the points of high dimension. In addition, Pyramid-tree creates non-discriminatory indices because the data that is located in the same pyramid slice has the same index value [174,175].

Recently, Zäschke et al. [176] proposed the structure PATRICIA-Hypercube-tree (PH-tree) based on the binary representation of data objects as a bit string [177] and the Quadtree structure [159], which uses hypercube for space partitioning in all dimensions at each node in the tree [178]. This partitioning allows to navigate more efficiently to the sub-node and stored entries more efficient compared to the binary trees [176]. Other improvements have been proposed to improve the efficiency of the PH-tree structure [179–181]. In [181], Favre Bully added new additional functions for data pre-processing and in [179], Bogdan Aurel proposed a new distributed architecture of the PH tree for parallel processing and cluster computing. However, consistency issues and the support of ACID properties (atomicity, consistency, isolation and durability) of transactions are not investigated [182].Furthermore, Costa et al. [183] proposed the ND-tree structure (Norm Diagonal Tree) to create a multidimensional indexing structure for high-dimensional data. It is based on a new data dimension reduction technique that uses the dual metric system, and the Euclidean standard and distance to support high-dimensional data (>100 dimensions) such as multimedia data. This technique reduces data dimension in two dimensions (2D) where they are indexed in the Quadtree tree which is considered a better dynamic indexing structure for two-dimensional data. In this approach, the reduction method applied to indexed data inevitably causes a loss of information on the original data, which reduces the precision of the search.

In the field of vehicle Internet (IoV), traffic management applications require efficient processing of requests with consideration of the massive trajectory data collected by the vehicles movement tracking process. For this purpose, Zhang et al. [153] proposed an online index system for vehicle trajectory data called PaIndex. The structure of the proposed index is based on multi-level partitioning of the space. At first the space is partitioned into regular grid cells in which the spatial domain of longitude and latitude is uniformly divided, then each cell's data is indexed in a hierarchical structure as $B^+$-tree. This partitioning allows to parallelize the insertion operations and the search requests to reduce the time and cost. Seismic data processing applications use the requests of the multidimensional range and to accelerate the processing of these types of request, Wang et al. [154] proposed an extension of the $B^+$-tree called MLB$^+$-tree index (Multi-level B$^+$-tree). MLB$^+$-tree is organized in several levels where each level contains several independent B$^+$-tree trees that allow the insertion and request to be performed in parallel after the top level. B$^+$-tree faces problems

of complexity, loss of space and consumption of many computational resources in massive data due to sub-optimal partitioning of nodes.

Jo et al. [184,185] proposed the Quadrant based Minimum Bounding Rectangle (QbMBR)-tree structure for processing large scale spatial data in HBase systems for an efficient processing, to reduce storage space and false positives in spatial query processing. The structure proposed in this work, partitions the space recursively into quadrants and for each quadrant, an MBR is created to provide secondary indexes that are stored in the HBase table. The recursive partitioning is terminated until the number of objects in MBRs is less than the partition threshold. Skip-octree is a new multidimensional data index in a cloud environment proposed by Dong et al. [186]. Skip-octree is based on two-level architecture that adapts the skip-list to accelerate the search process and octree structure is used in each server to store and index multidimensional data in a hierarchical way. A new indexing technique was proposed by Malhotra et al. [187] called SkipNet-Octree based on the combination of two structures SkipNet [188] and compressed Octree [186] to index and process queries on multidimensional data in Cloud Computing. SkipNet-Octree is a two-layer structure where the top layer represents a global index created through the SkipNet structure that contains metadata for local index nodes and the Octree index technique used to create a local index [187]. The experiments carried out show that the SkipNet-Octree technique works better than traditional Skiplist and Octree for complex queries.

A new hybrid multidimensional data indexing structure was presented in [189]. The structure is based on the concepts of grid, pyramid, and height to partition space and design the key to effectively access data. In this structure, the space is partitioned into grids, and each subspace (grid cell) is identified by pyramids and heights [189,190]. The main objective of this hybrid structure is to create a structure that supports floating-point numbers and reduces the number of I/Os to ensure high system throughput and more efficient execution of range requests [189].

Recently, Samson et al. [155] proposed a new static spatial data indexing structure called aX-tree (Packing X-tree) to avoid performance degradation for high-dimensional databases encountered by the X-tree [131] structure and their variances (X$^+$-tree [156], VA-File [162] etc.). aX-tree uses the Bulk-Loading technique to reduce the amount of empty space, fast loading and better partitioning of space based on MBR. With this technique, aX-tree has overcome the over expansion of the super-node where it became a structure characterized by: (i) minimum tree height (ii) high directory node quality (iii) minimum overlap and (iv) reduced area the MBR and most importantly, maximized space efficiency [155].

Sprenger et al. in [157,191] introduced BB-tree. It is a new multidimensional index structure that combines the Kd-tree [158] and X-tree [131] structures. BB-tree is a quasi-balanced tree, supports complete- and partial-match range queries, exact-match queries, and dynamic updates. The authors created this structure based on recursive partitioning of the space into k partitions as for Kd-tree. BB-tree is based on the structure of elastic bubble buckets in the leaf nodes of the tree like the X-tree. These buckets store data (subset) to balance the structure. The leaf nodes (or regular BB) has a limited capacity ($b - max$). The latter is transformed into super-nodes (super BB) similar to the structure of X-tree in case of saturation; the data from these nodes is scanned linearly. According to the results of the experiments [86], BB-tree shows a better efficiency for range queries compared to R*-tree [138], Kd-tree, PH-tree [176] and VA-file [162] but Knn similarity search queries are not taken into account in this structure. Tables 9 and 10 present, respectively, an analytical and comparative study of multidimensional indexing techniques based on space partitioning.

**Table 9.** Analysis of multidimensional indexing techniques based on space partitioning.

| Proposition | Refs | Dataset Type | Data Dimension | Indexing Nature | Complexity (Estimation) | |
|---|---|---|---|---|---|---|
| | | | | | Insertion and Deletion | Search |
| Kd-tree | [158] | Geo-graphical | | | $O(dnlog(n))$ | $O(n\log(n))$ |
| Adaptive Kd-tree | [167] | Files | | | $O(dn\log(n))$ | $O(n\log(n))$ |
| KdB-tree | [168] | Floating point numbers | | | $O(n\log(n))$ | $O(n(\frac{k-1}{k}))$ |
| SKd-tree | [169] | Spatial | | | Not estimated | Not estimated |
| Quad-tree | [159,160] | Spatial | Multidimensio-nal | Dynamic | $O((d+1)n\log(n))$ | $O(n\log(n))$ |
| PH-tree | [176] | Synthetic | | | $O(n\log(n))$ | $O(n\log(n))$ |
| ND-tree | [183] | Synthetic | | | Not estimated | Not estimated |
| QbMBR-tree | [184,185] | Synthetic, spatial | | | Not estimated | Not estimated |
| VA-file | [162] | Synthetic data and images | | | Not estimated | Not estimated |
| Octree | [186] | Spatial | | | $O((d+1)n\log(n))$ | $O(n\log_8(n))$ |
| Pyramid | [161] | Synthetic data | | | Not estimated | Not estimated |

**Table 10.** Summary of advantage and disadvantage of multidimensional indexing techniques based on space partitioning.

| Proposition | Refs | Advantages | Disadvantages | |
|---|---|---|---|---|
| Kd-tree | [158] | • Balanced hierarchical split<br>• Simple implementation | • Costly and arbitrary<br>• Low use of allocated space<br>• Performance limited by data dimension | |
| Adaptive Kd-tree | [167] | • Lower-cost k-nn research<br>• Storage at leaf nodes | • Not appropriate for frequent insertion and deletion | |
| KdB-tree | [168] | • Height-balanced structure<br>• Efficient search for point queries | • Supports only point data<br>• Cannot guarantee minimum storage utilization<br>• Insufficient research performance | |
| SKd-tree | [169] | • Suitable for non-zero size spatial objects<br>• Ensures good storage | • Slow performance even in high dimensional spaces | |
| Quad-tree | [159,160] | • Efficient storage and retrieval | • Not balanced structure<br>• Does not consider the spatial distribution of the data during the partitioning phase<br>• Not suitable for higher-dimensional data | • Degradation on large scale |
| PH-tree | [176] | • Faster and more efficient in terms of space efficiency, query and update performance | • Supports point and range query only<br>• High memory consumption | |
| ND-tree | [183] | • Support high-dimensional data | • Loss of information on the original data<br>• Search performance limited by data dimensions | |
| QbMBR-tree | [184,185] | • Reduce the false positives in spatial query<br>• Reduces the storage space<br>• Reduce query execution times | • Overlap of MBRs | |
| VA-file | [162] | • Simple implementation<br>• Sequential search improved | • Large dimension heavy coding<br>• Degradation on large scale | |
| Octree | [186] | • Better spatial management and k-nn search | • Support only 3-dimensional data | |
| Pyramid | [161] | • Degradation on large scale<br>• Linear increase of cells | • Poor request processing k-nn<br>• Degradation on large scale | |

### 5.1.3. Bitmap-Based Technique

Bitmap index (also known as BitArray or vector-based index) is an efficient indexing structure for search and retrieval of large databases and data warehouses (DW) with less complexity and is very efficient when attributes have a low number of distinct values. This technique is used by several popular commercial systems such as Oracle [192,193] and SybaseIQ [194,195]. Bitmap index technique is based on the representation of the existence or absence of a specific property by a sequence of bits where each bit (0/1) represents the value of an attribute for a given tuple such that the bit sequence has a 1 in position $i$ if the $i^{th}$ data element meets the property, and 0 otherwise [196,197]. Bitmap index uses logical operations, such as AND, OR, NOT and XOR to respond and accelerate responses to complex queries [198].

Traditional bitmaps are suffering from a problem of space over-consumption, especially for highly cardinal data. To address this challenge and for faster retrieval, compressed bitmap indexes are recommended. As a consequence, many efficient bitmap compression algorithms have been developed, including: BBC (Byte-aligned Bitmap Compression) [192], WAH (*Word-Aligned Hybrid*) [198,199], PLWAH (*Position ListWAH*) [200], EWAH (*Enhanced Word-Aligned Hybrid*) [201], CONCISE (*Compressed N Composable Integer Set*) [202], VALWAH (*Variable-Aligned Length WAH*) [203], SECOMPAX (*Scope-Extended COMPressed Adaptive indeX*) [204], SBH (*Super Byte-aligned Hybrid*) [205], Roaring [206], SPLWAH (*PLWAH algorithm for sorted data*) [207], BAH (*Byte Aligned Hybrid compression coding*) [208], cSHB (*Compressed Spatial Hierarchical Bitmap*) [209] and CODIS (*COmpressing DIrty Snippet*) [210]. Through these compression algorithms with logical operations, the execution time is reduced compared to the basic bitmap index without compression, which is an essential property of bitmap indexing [211].

Recently, Chenxing et al. [208] proposed a new compression algorithm more similar to WAH algorithm named BAH (Byte Aligned Hybrid compression coding) whose objective is to improve the performance in terms of space and the efficiency of the requests. BAH uses simple rules for raw bitmap encoding compared to other WAH variants that use a more complicated code book. BAH uses SIMD operations to accelerate the efficiency of the AND operation on multiple compressed bitmaps. Another compression algorithm has been proposed based on the WAH algorithm called CODIS which is proposed by Wenxun et al. [210]. The basic idea of CODIS is the reduction of space through the representation of the bit string in the bitmap index with fewer bits, without influencing the efficiency of the index. The results obtained during the experimentation demonstrate that this technique is more efficient than the other algorithms in the literature, including WAH [199], COMPAX (COMPressed Adaptive indeX) [212] and PLWAH [207].

Bitmap index method is a very efficient technique for answering complex queries for read-only systems and for data that is not frequently updated as a data warehouse, but it is less efficient in other cases (i.e., for data that is frequently updated). This problem is caused by the compression process where the latter is used to reduce the storage space (as we said before), but at each update operation, it is necessary to decode and encode the bitmap, and this operation is very expensive [213]. Manos et al. [213] proposed a new bitmap index named UpBit (Updatable Bitmap) to overcome this problem. This index offers efficient updates without affecting read performance. The UpBit index adds an additional update vector for each bitmap vector in which update processes will be performed on the update vector where the latter stores updates corresponding to its value bitmap only. Bitmap minimizes the cost of decoding as well as improves navigation through the use of closing pointers on bit vectors. Chigullapally et al. [214] proposed an extension of this structure, where the authors parallelized the merging of bit vectors to improve the performance of the UpBit index.

*5.2. Metric Indexing Techniques*

Formally, a metric space $(\mathcal{O}, d)$ is a set of points $\mathcal{O}$ (where, $\mathcal{O} \neq \varnothing$) to which we associate a notion of distance $d(\mathcal{O} \times \mathcal{O}) \rightarrow \mathbb{R}^+$ between the elements that meets the following properties:

- *Non-negativity* : $\forall (x, y) \in \mathcal{O}^2, d(x, y) \geq 0$;
- *Identity* : $\forall x \in \mathcal{O}, d(x, x) = 0$;
- *Symmetry* : $\forall (x, y) \in \mathcal{O}^2, d(x, y) = d(y, x)$;
- *Triangle inequality* : $\forall (x, y, z) \in \mathcal{O}^3, d(x, y) + d(y, z) \leq d(x, z)$.

**Partitioning of Space:** In the literature, two partitioning techniques have been developed: the first technique is based on hyper-sphere (or ball) partitioning as in: VP-tree [215], mVP-tree [216] and MM-tree [217], etc. while the second technique is based on hyper-plane partitioning as in: GH-tree [218], GNAT-tree [55] and EGNAT-tree [219], etc.

**Hyper-sphere (or Ball) partitioning**: VP-tree [215] is a hierarchical indexing structure such as Kd-tree, developed to improve the search for similarity in a metric space. VP-tree is a technique based on the partitioning of the space through the balls according to distance. VP-tree uses the median distance between the vantage point (choose randomly) and the points of the space to partition the space in two balanced disjoint sub-spaces. The disadvantage of the VP-tree is the highest cost in terms of calculated distance as well as time, especially as the data space has a large dimension where the number of branches searched for is high [220]. The mVP-tree is proposed to address the problem of reduced performance in the search for similarity of the VP tree in high-dimensional metric spaces. mVP-tree (multiple Vantage Points tree) [216] is an extension of the VP-tree idea that uses several vantage points instead of one. The major advantage of mVP-tree over VP-tree is that it also uses pre-computed distances (at the construction step) to improve search speed and reduce the number of distance calculations as well as the time required to execute queries. The experiments presented in [215] show that mVP-tree improves the VP-tree slightly better but not in all cases, while a greater improvement is achieved when several pivots per node are used [61].

Cheng et al. [221] proposed the DMVP-tree structure to accelerate the recovery process of similarity images in the airport's video surveillance system. This approach is an improvement of the metric indexing structure MVP-Tree [216] using the horizontal distribution of MVP-Tree structure in several machines to overcome massive high-dimensional spatial indexing problems. The DMVP-tree structure partitions the space horizontally where the upper area is called "main space" and the lower areas are called "secondary spaces". The main space is indexed in MVP-Tree that stored in the master machine and the secondary space is partitioned in a static way on the slave machines.

MM-tree [217] is another indexing structure for metric space that also uses the principle of recursive partitioning of space through balls in two non-overlapping regions. MM-tree is an unbalanced structure due to the different size of sub-spaces (or regions), due to the external region of the balls. To solve this problem, MM-tree applies an additional a semi-balanced algorithm that allows to re-organize the objects of the leaf nodes. According to the experiences presented by [222], MM-tree does not support high-dimensional data. Several structures have been proposed based on the MM-tree structure. All these structures are developed to address the challenges of the MM-tree structure (Onion-tree [222], IM-tree [223] and XM-tree, the extended Metric tree [224]).

Onion-tree [222] is an MM-tree improvement proposed to overcome the challenges or limitations of the MM-tree structure. The Onion-tree is very fast to respond the similarity search requests thanks to the increase in the number of partitions of the space compared to the MM-tree, but the problem with this structure remains in the very slow building because of the re-insertion of objects. IM-tree [223] is a proposed structure to address the problem of index degeneration posed by the fourth region of MM-tree and onion-tree. IM-tree selects the two most distant points as pivots and splits the fourth region in two using a plane. For massive data, the external region of the balls of the IM-tree becomes very large, which can then lead to the degeneration of the index.

XM-tree [224] is an extension of the IM-tree [223] structure that is based on the successive division of space with spheres. XM-tree is proposed to address the problem of degeneration of the index mentioned above by focusing on minimizing the size of the outer regions of the balls. To achieve this goal, XM-tree creates extended regions inspired by the X-tree [131]. The extended regions make the Knn search very fast thanks to the elimination of some objects that are not necessary to calculate the relative distances of a query object.

With the same principle of IM-tree, the NOBH-tree (Non-Overlapping Balls and Hyper-planes tree) [225] partitions the metric space through the hyper-planes and hyper-spheres to organize the data into non-overlapping regions as well as to reduce the number of distance calculations required to answer the questions. The NOBH tree recursively divides the space into several regions using the pivots $(p_1, p_2)$, and separates the data such that the distance evaluation of an element $X_i$ at $p_1$ and $p_2$ can only contain the region $X_i$. These regions

are divided using a metric hyper-plane and two spheres, where the radius of the sphere $r$ corresponds to the distance between $p_1$ and $p_2$. The main drawback of this technique is the complexity of the enclosing forms. This increases the cost of insertion and search operations [224].

Ball*-tree [226] is a binary tree more balanced where each node defines a D-dimensional hyper-sphere, or a ball, that contains a subset of the points to be searched. Ball*-tree is an improvement of the original structure of Ball-tree [227,228] proposed by Dolatshah et al. in [226]. Ball*-tree addresses the problem of data distribution and the unbalanced structure of Ball-tree by taking into account the data distribution when determining the splitting hyper-plane. In Ball*-tree the splitting hyper-plane is perpendicular to the first principal component using principal component analysis (PCA). Using this splitting technique allows to create a more balanced and efficient tree structure unlike the Ball-tree where the splitting hyper-plane is determined by the line that connects the two furthest points which creates unbalanced sub-partitions.

A new indexing structure for indexing IoT data called BCCF-tree (Binary Container-based Binary Tree in Cloud-Fog Computing) was introduced by Benrazek et al. [229]. BCCF-tree uses the k-means algorithm to partition the space into subspace without overlapping. This structure is adapted to the cloud-fog computing architecture. The aim is to improve the quality of the discovery and recovery process of large IoT data by sharing the system load among the elements of the cloud-fog computing architecture. According to the results presented in this work, the BCCF-tree shows a good performance in terms of similarity search. However, the results also show that the construction of this structure is very expensive in terms of complexity, where it reached $O((n \cdot \log n) \cdot 2(t \cdot n))$ [230,231]. Tables 11 and 12 present, respectively, an analytical and comparative review of metric indexing techniques based on ball partitioning.

**Table 11.** Analysis of metric indexing techniques based on ball partitioning.

| Proposition | Refs | Dataset Type | Data Dimension | Indexing Nature | Complexity (Estimation) | |
|---|---|---|---|---|---|---|
| | | | | | **Insertion and Deletion** | **Search** |
| VP-tree | [215] | Images | Dynamic | | $O(n \log_2(n))$ | $O(n^2 \log(n))$ |
| mVP-tree | [216] | Images | Static | | $O(n \log_m(n))$ | $O(mn \log(n))$ |
| MM-tree | [217] | Image and Geographic coordinates | | | $O(n^2 \log_4(n))$ | $O(n^2 \log(n))$ |
| Onion-tree | [222] | Image, Time-series and Geographic coordinates | | Multidimensional | $O(n^2 \log(n))$ | $O(n^2 \log(n))$ |
| IM-tree | [223] | Image | | | $O(n \log(n))$ | $O(c_{max} \log(n))$ |
| XM-tree | [224] | Geographic coordinates and Image | Dynamic | | $O(nmx \log(n))$ | $O(c_{max} \log(n))$ |
| Ball-tree | [227,228] | Not mentioned | | | $O(n \log(n))$ | $O(d \log(n))$ |
| Ball*-tree | [226] | Synthetic and Point data | | | $O(n \log(n))$ | $O(\frac{n}{d} \log(n))$ |
| NOBH-tree | [225] | Image and Synthetic | | | $O(n \log_m(n))$ | $O(n \log(n))$ |
| BCCF-tree | [229] | Synthetic, Geographic coordinates and wearable action recognition database | | | $O((n \log n) \cdot 2(tn))$ | $O(\frac{1}{2}\sqrt{n} \log_2(k) + (log(n)/(\frac{1}{2}\sqrt{n}))k)$ |

**Table 12.** Summary of advantage and disadvantage of metric indexing techniques based on ball partitioning.

| Proposition | Refs | Advantages | Disadvantages | |
|---|---|---|---|---|
| VP-tree | [215] | • Simple implementation | • Highest distance and time<br>• Research costs increase in large dimensions | |
| mVP-tree | [216] | • Reduces research costs<br>• Little affected on a large e scale | • Static structure<br>• Support only range research | • Degradation on large scale |
| MM-tree | [217] | • Best space partitioning<br>• Non-overlapping regions | • Degeneration of the index (fourth region) | |
| Onion-tree | [222] | • Better partitioning of space | • «Reinsertion» objects (semi-balancing) | |
| IM-tree | [223] | • Efficient compared to MM-tree and Slim-tree | • Index degeneration in massive data | |
| XM-tree | [224] | • Minimize the size of the search regions<br>• Fast k-nn search | • Requires high memory space | |
| Ball-tree | [227] | • Efficient brute force search in large dimensions | • Unbalanced structure<br>• Longer build times | • The problem of overlap not effectively addressed |
| Ball*-tree | [226] | • More balanced and efficient structure compared to Ball-tree | • Performance decreases as the dimensionality of the data increases | |
| NOBH-tree | [225] | • Non-overlapping division of the data space | • High cost of insertion and research | |
| BCCF-tree | [229] | • Non-overlapping division of the data space<br>• Fast k-nn search<br>• Balanced data partitioning | • Expensive construction | |

**Hyper-plane partitioning**: The first indexing structures that are based on the partition of space through hyper-planes are the oldest structure BS-tree (Bisector tree) [232] and the structure GH-tree (Generalized Hyper-plane tree) [218] which is similar to BS-tree. GH-tree is a binary indexing structure that divides the space recursively into two sub-spaces through the hyper-plane which is defined by the two representative points or pivots (the two farthest points as in [233–235]) the rest of the points are partitioned according to the distance between these pivots. The drawbacks of this structure reside in the search process where at each node two distance operations are performed which increases the cost of the search, as well as the selected pivots does not guarantee the best partition of space, which makes the problem of index degeneration possible. GNAT-tree (Geometric Near-neighbor Access Tree) [55] is a static indexing structure. GNAT-tree is a generalization of the GH-tree which uses $m$ pivots in each internal node instead of two (i.e., GNAT-tree is an $m$-ary tree). Regarding EGNAT-tree [219], it is the dynamic structure of GNAT-tree.

Recently, GHB-tree (Generalised Hyper-plane Bucketed) [236] is proposed as an improvement of the GH-tree structure. The objective of the GHB-tree structure is created a balanced indexing structure with less construction cost through the new type of node that they called a bucket. These types of nodes found at the leaf level which has a limited capacity to store a subset of the most similar data to improve the search process. The CD-Tree, cited in [237], is a type of index based on hyper-plane partitioning. This indexing approach has proven effective for a limited number of dimensions but remains ineffective for large dimensions. The recursive partitioning of space into two regions is the principle of this technique. Two pivots are chosen each time, and each one is associated with the closest objects. However, the problem with this technique is that the geometrical shapes of the regions pose many problems in the search algorithm.

A new metric indexing structure called SPB (Space filling curve and Pivot based B$^+$-tree) tree was proposed by Chen et al. [238,239]. The method was proposed to improve the efficiency of similarity search, support large number of complex objects and reduce the cost in terms of storage, construction and search (i.e., reduces CPU and I/O cost). To achieve these objectives, SPB-tree uses geometric information not available in metric space through the mapping of objects in a metric space to data points in a vector space using well-chosen pivots. The B$^+$-tree with MBB (Minimum Bounding Boxes) is used to index the one-dimensional data generated by the function of dimensionality reduction Space-Filling Curve (SFC) applied to the data points of the vector space. Although the structure is very

simple, but the construction steps such as space transformations and pre-treatment can be made parallelism is very difficult [240].

Compared between the two partitioning strategies (hyper-sphere and hyper-plane), it can be observed that the problem of node overlap is a problem that has not been effectively addressed by the techniques based on partitioning by hyper-sphere, but this problem does not exist in hyper-plane techniques (such as: GH-tree and GNAT). On the other, structures based on hyper plane partitioning are more difficult to maintain their balance because of the uncontrolled insertion positions of new elements [219]. A comparative and analytical study of metric indexing techniques based on hyper-plane partitioning is presented in Tables 13 and 14, respectively.

**Table 13.** Analysis of metric indexing techniques based on hyper-plane partitioning.

| Proposition | Refs | Dataset Type | Data Dimension | Indexing Nature | Complexity (Estimation) | |
| --- | --- | --- | --- | --- | --- | --- |
| | | | | | Insertion and Deletion | Search |
| *BS-tree* | [232] | Point data | | | $O(n \log_2(n))$ | not estimated |
| *GH-tree* | [218] | Not mentioned | | Static | $O(n \log_2(n))$ | $O(n \log(n))$ |
| *GNAT-tree* | [55] | Image, text, Vectors | | | $O(nm \log_m(n))$ | $O(\frac{n}{m} \log(n))$ |
| *EGNAT-tree* | [219] | Words and coordinate space | Multidimensional | | $O(nm \log_m(n))$ | $O(\frac{n}{nd} \log(n))$ |
| *GHB-tree* | [236] | Geographic coordinates and Image | | | $O(n \log(n))$ | $O(2k \log_2(k))$ |
| *CD-tree* | [237] | Image | | Dynamic | $O(n \log(n))$ | $O(n \log(n))$ |
| *SPB-tree* | [238,239] | Words, Colors, DNA, Signature and Synthetic | | | $O(nlx + nm \log_m(n))$ | not estimated |

**Table 14.** Summary of advantage and disadvantage of metric indexing techniques based on hyper-plane partitioning.

| Proposition | Refs | Advantages | Disadvantages | |
| --- | --- | --- | --- | --- |
| BS-tree | [232] | • Fast k-nn search and orthogonal queries | • Requiring linear space | • Degradation on large scale |
| GH-tree | [218] | • Simple partitioning Reduced overlap rate | • Complicated form to manipulate<br>• Degeneration of the index<br>• High cost search | |
| GNAT-tree | [55] | • Non-overlapping<br>• Improve the search | • Static and complicated structure<br>• High computational costs | • More difficult to maintain index balance |
| EGNAT-tree | [219] | • Non-overlapping<br>• Requires less CPU time than the GNAT-tree | • Degradation on large scale<br>• More difficult to maintain index balance | |
| GHB-tree | [236] | • Balanced structure | | |
| CD-tree | [237] | • Efficient re-construction time | • Ineffective in large dimensions<br>• Ineffective search | • Degradation on large scale |
| SPB-tree | [238] | • Simple structure<br>• Reduce the cost in terms of storage, construction and search<br>• Effective similarity search | • Difficult to parallelize it | • More difficult to maintain index balance |

**No Partitioning of Space (Partitioning Data):** This category does not require space partitioning. Among the families that use this type of partitioning (partitioning data), we find essentially the M-tree family. M-tree [241] is a metric tree structure height-balanced allowing incremental updates based on the grouping of dynamic data into balls (or hyper-spheres). M-tree stores some data in internal or inner nodes for routing purposes and the remainder is stored in the leaf nodes. M-tree suffers from the problem of overlapping sub-spaces, which increases the number of distance calculations to answer a query [224,242]. Several recent structures share the same principles of the M-tree and Slim-tree [243] is one of them. Slim-tree improves the structure of the M-tree with a new splitting technique based on the minimum spanning tree (MST). Slim-tree also reduces the cost of construction in addition to this, it introduces a post-processing method which reduces overlap and, consequently, the cost of research. The major disadvantage of this algorithm is the ability to generate nodes with few objects and/or empty nodes, which significantly reduces the performance of the index, especially in large spaces [244–246].

The work of Murgante et al. [247] aims to avoid unsatisfactory node partitioning and reduce regional overlap in the M-tree structure [241,248]. The authors proposed a new metric indexing structure called $M^X$-tree based on the original M-tree structure. $M^X$-tree implements the concept of super-nodes inspired by the [131] structure of the X-tree. This structure avoids the unsatisfactory division of nodes, thus reducing the cost of computation and extends it completely to metric space where temporal complexity is reduced to $O(n^2)$ without setting any parameter. As for the M tree, the temporal complexity reaches $O(n^3)$. The authors also add another strategy to the $M^X$-tree structure to improve the management of free memory space that is represented in the indexing of tree leaf objects in an internal index [249] through the vantage-point tree (VP-tree) [215]. Due to the symmetry of the metric axioms of metric space, metric indexing techniques such as M-tree and their variances cannot answer to the approximate requests of sub-sequences or subsets. Bachmann [250] propose an improvement on the M-tree structure called SuperM-tree to create a metric indexing structure capable of responding to the approximate requests of sub-sequences or subsets such as searching for a similar partial sequence of a gene, a similar scene in a film, or a similar object in an image. The author introduces a new metric measurement subset space "Metric Subset Space (M; d; v)" to create this structure. It ignores the symmetry of metric axioms and adds a new relationship on object size (for more details on the demonstration of this new space, see article [250]).

The efficiency of the search in M-tree is reduced when the volume is high, thus, Pivoting M-tree (PM-tree) is proposed [251,252] to resolve this problem. PM-tree is a hybrid structure, which combines the "local-pivoting strategies" of M-tree [241] with the "global-pivoting strategies" of LAESA [253]. Recently, Razent et al. [254] presented a new construction algorithm for the two indexing structures M-tree and PM-tree. The objective is to enhance the performance of Knn requests. The construction algorithm is based on storing data once in the tree (M-tree or PM-tree) through the deletion of promoted elements that are stored in the upper level of the leaf nodes during their partitioning. To achieve this idea, the authors use the aggregate nearest query to find the most efficient local pivots that will be promoted during the partitioning of internal nodes. According to the report of the experiments carried out in [254], this algorithm reduces node occupancy, reduces overlap between nodes and increases significantly the performance of search operations in terms of speed compared to the construction algorithm of the M-tree and PM-tree structure.

Navarro et al. [255] proposed the DSC (Dynamic Set of Clusters) structure, a new dynamic metric index structure that reduces memory consumption. DSC is a combination of two new structures proposed in [255]. The first structure is a hierarchical structure called DSAT (Dynamic Spatial Approximation Tree). This structure uses timestamps that indicate the moment when elements were inserted to avoid the reconstruction of the structure after updates as well as in the pruning process of similarity queries [256]. The second structure is a variant of the List of Clusters (LCs) structure called Dynamic List of Clusters (DLCs). DLC is a secondary memory-based structure in which it reduces memory consumption compared to the original LC [257] where the M-tree structure is used as a partitioning technique. DSC is a structure divided into two parts. A part stored in the main memory as DSAT structure and the second part stored in the disk which represents by the DLC structure [255].

Through the MapReduce framework, Chanet et al. [258] proposed two partitioning techniques for joins of metric similarity to balance the load [259]. The first method focuses on the selection of centroids and clustering data in a one-dimensional space through the Space-Filing-Curve (SFC) technique. This technique allows to partition the data in equal size thanks to the high quality of the selected centroids. The second partitioning method based on the Kd-tree structure [158,260], which divides the data after the pivot mapping [258].

Because of missing data generated by different application areas, indexing structures are distorted where the latter produces a bias in the response to the query. To solve this problem, Brinis et al. [261] proposed the Hollow-tree structure, which enables missing

data to be managed without distracting from its structure. Hollow-tree is a metric access method that uses the CFMLI (Complete First and Missing Last Insert) technique to provide a strategy for building metric indices. This strategy consists of indexing all complete data in the first steps to create a coherent structure then insert the elements with the missing values (with NULLS) at the nodes of the sheets. All this is achieved by the ObAD (Observed Attribute Distance) technique, which makes it possible to compare elements with missing values based on distance functions.

Yang et al. in [262] proposed an Asynchronous Metric Distributed System (AMDS) for metric spaces to process metric similarity requests efficiently in a distributed environment. In the proposed system, the authors adapt the technique of pivot mapping, which enables to divide the data uniformly into non-joint fragments and provide load balancing. To reduce computation costs in the process of similarity research, the Minimum Bounding Box (MBB) technique is used. The AMDS system supports large-scale similarity requests in metric spaces simultaneously through synchronous processing based on publication/subscription communication mode. Tables 15 and 16 analyze and compare metric indexing techniques based on data partitioning.

**Table 15.** Analysis of metric indexing techniques based on data partitioning.

| Proposition | Refs | Dataset Type | Data Dimension | Indexing Nature | Complexity (Estimation) | |
|---|---|---|---|---|---|---|
| | | | | | **Insertion and Deletion** | **Search** |
| M-tree | [241] | Synthetic data | | | $O(mn \log_m(n))$ | $O(mn \log(n))$ |
| Slim-tree | [243] | Spatial, Face vectors and Text | | | $O(n^2 \log(n))$ | $O(n^2 \log(n))$ |
| Slim*-tree | [263] | Image and Spatial | | | $O(n^2 \log(n))$ | $O(\frac{n^2}{d} \log(n))$ |
| MX-tree | [247] | Image, Text | | | $O(n^2 \log(n)) + n^2$ | $O(n^2 \log(n))$ |
| SuperM-tree | [250] | Synthetic data | Multidimensional | Dynamic | not estimated | not estimated |
| PM-tree | [251,252] | Synthetic data | | | $O(n(m + l) \log_m(n))$ | $O(n^2 log(n))$ |
| DSC | [255] | Vectors, Text and Colors | | | not estimated | not estimated |
| SFC & Kd-tree | [258] | Synthetic, Text, DNA and Color | | | $O(n \log(n))$ | $O(kn \log(n))$ |
| Hollow-tree | [261] | Synthetic | | | not estimated | not estimated |

**Table 16.** Summary of advantage and disadvantage of metric indexing techniques based on data partitioning.

| Proposition | Refs | Advantages | Disadvantages | |
|---|---|---|---|---|
| M-tree | [241] | • Balanced height structure<br>• Reduction of distance calculations | • Problem of overlaps<br>• High cost search<br>• No adapted to highly grouped data | |
| Slim-tree | [243] | • Efficient compared to M-tree<br>• Reduced overlap rate | • The overall computational complexity | |
| Slim*-tree | [263] | • Reduces the cost of calculation during reconstructing<br>• Avoids the unsatisfactory division | • Reinserting objects is largely costly | |
| MX-tree | [247] | • Reduces the cost of calculation during reconstructing<br>• Avoids the unsatisfactory division | • High cost search | |
| SuperM-tree | [250] | • Capable of responding to approximate requests for subsequences or subsets | • Expensive construction<br>• Evaluates only for research 1-nn | • Degradation on large scale |
| PM-tree | [251] | • More efficient similarity search compared to M-tree | • Not support the k-nn search<br>• Expensive construction compared to M-tree | |
| DSC | [255] | • Reduces memory consumption | • High amount of distance calculations | |
| SFC & Kd-tree | [258] | • High quality of the selected centroids<br>• Effective partitioning<br>• Better query performance | • Not support the k-nn search | |
| Hollow-tree | [261] | • Capable of managing missing data | • Lower accuracy in small data | |

At the end of this section, Table 17 presents the indexing techniques and related applications discussed earlier.

**Table 17.** Application area of the indexing structures.

| Indexing Structure | Application |
|---|---|
| LSH | • Pattern matching<br>• Recommendation retrieval<br>• Text processing<br>• Natural language processing<br>• Reducing the dimensionality of data<br>• Image/Video retrieval<br>• Content similarity deployment and discovery |
| Kernelized LSH | • Content-based retrieval<br>• Speaker search<br>• Image classification |
| Robust Discrete Spectral Hashin | • Image semantic indexing<br>• Image retrieval |
| Spectral Hashing | • Image retrieval<br>• Detection of region-duplication forgery in digital images<br>• Fast approximate nearest neighbor<br>• Classification |
| Kernel Based Supervised Hashing | • Person re-identification<br>• Similarity search<br>• Image retrieval |
| Label-regularized Max-margin Partition | • Classification for large-scale datasets |
| Bit-Scalable Deep Hashin | • Similarity learning for image retrieval and person re-identification |
| Asymmetric Deep Supervised Hashing | • Image retrieval |
| M-tree | • Similarity search in multimedia Dataset<br>• Accelerator for database query<br>• Recommendation System<br>• Indexing the music data<br>• Classification |
| Slim-tree | • Video indexing and similarity search |
| SFC & Kd-tree | • Data cleaning and data mining |
| Hollow-tree | • Store and retrieve large volumes of complex data |
| AMDS | • Multimedia retrieval<br>• Computational biology<br>• Location-based services |
| VP-tree | • Pattern recognition and image processing<br>• Image indexing and retrieval<br>• Storing neuronal morphology data<br>• Similarity search on cloud computing<br>• Malware detection<br>• Clustering |
| mVP-tree | • Images retrieval in airport video monitoring systems |
| MM-tree | • Image retrieval |
| XM-tree | • Web Information Retrieval |
| Ball-tree | • Face sketch recognition<br>• Classification in high dimensions<br>• Clustering and matching for object class recognition |
| BCCF-tree | • Image indexing and retrieval for person re-identification<br>• Indexing IoT sensor data |
| GH-tree | • Image Search by Content |
| GNAT-tree | • Indexing and similarity search of face-images data |
| SPB-tree | • Multimedia retrieval<br>• Pattern recognition<br>• Computational biology |
| X-tree | • Image coding<br>• Classification |
| Kd-tree | • Search and synchronization of sensor nodes |
| R-tree | • Classification<br>• Spatial indexing for the IoT data management<br>• Images search and retriever<br>• Geographical search |
| Hilbert R-tree | • Visualization of 3D massive data |

## 6. A Comparative Analysis of Multidimensional Indexing Methods

This section presents several algorithms that were initially proposed for spatial indexing structures. The similarity research applications usually use vectors to describe the data; vectors can be obtained by extracting (domain-specific) features. Using multi-dimensional indexing structures, these feature vectors are indexed, and since they apply a form of spatial indexing, a search tree can be defined to perform similarity (or proximity) queries on all vectors. This section provides an overview of the different methods of multidimensional access developed over the last two decades and compares their performance. The variety of data structures and their experimental performance give a fairly accurate idea of their advantages and disadvantages. However, the performance of a particular data structure depends on many factors such as the hardware used, operating system settings, buffer sizes, page sizes and datasets. Furthermore, performance is usually measured in terms of the number of disk accesses, search time, etc.

However, several researchers have argued that no single method of access has been found to be far superior to all others [62,263–266]. While one experimental result declares a structure to be the final winner, a different experimental result may be the same or inferior. The reason why these comparisons are so difficult is the number of different criteria used to define it as optimal. A summary of the interesting and missing elements of each proposal is presented at the end of the literature review. It includes a synthesis of all the properties of indexing techniques based on the non-division of multidimensional space. It also provides an overview of the use of encompassing geometric forms that allow for a more refined filtering of regions in the search phase.

In the first part, an approach based on the non-partitioning of space was presented. The well-known R-tree technique begins the application of the principle of interlocking shapes with the creation of hierarchically interlocking hyper-rectangles. Unfortunately, this method suffers from the problem of the curse of the dimension: inefficiency in large dimensions. In the same context, the $R^*$-tree is based on the principle of the reinsertion of objects to minimize the recovery rate between forms. Thus, the $R^*$-tree proposes to reinsert the saturated page (node) into the same level of the tree before splitting it. In most cases, this reintegration allows to avoid splitting and it ensures a continuous reorganization of the tree. Then, the -tree technique was introduced by creating super knots (refusing to create a hierarchy with too many overlaps and adopting a local strategy of extensive storage). It manages collections much better. However, unfortunately, if the size increases, this technique loses its value. Another approach, the SR-tree, is based on the intersection shape between rectangles and spheres. The problem with this technique is the complexity of the encompassing forms, which increases the cost of insertion operations and searches.

Furthermore, another type of approach was introduced, based on the partitioning of multidimensional space, such as the *k*D-tree. The principle of spatial partitioning eliminates the problem of overlapping shapes. In this type of strategy, a problem exists when a demand point is near the boundary between two regions. Therefore, it is necessary to visit all neighbouring regions. On the other hand, the procedure for splitting a saturated page does not depend on the spatial distribution of the data.

For more details, recall Figure 8 which presents a taxonomy, Tables 7–10 and 17 which provide an in-depth analysis, summarize the advantages and disadvantages, and application areas of multidimensional indexing techniques.

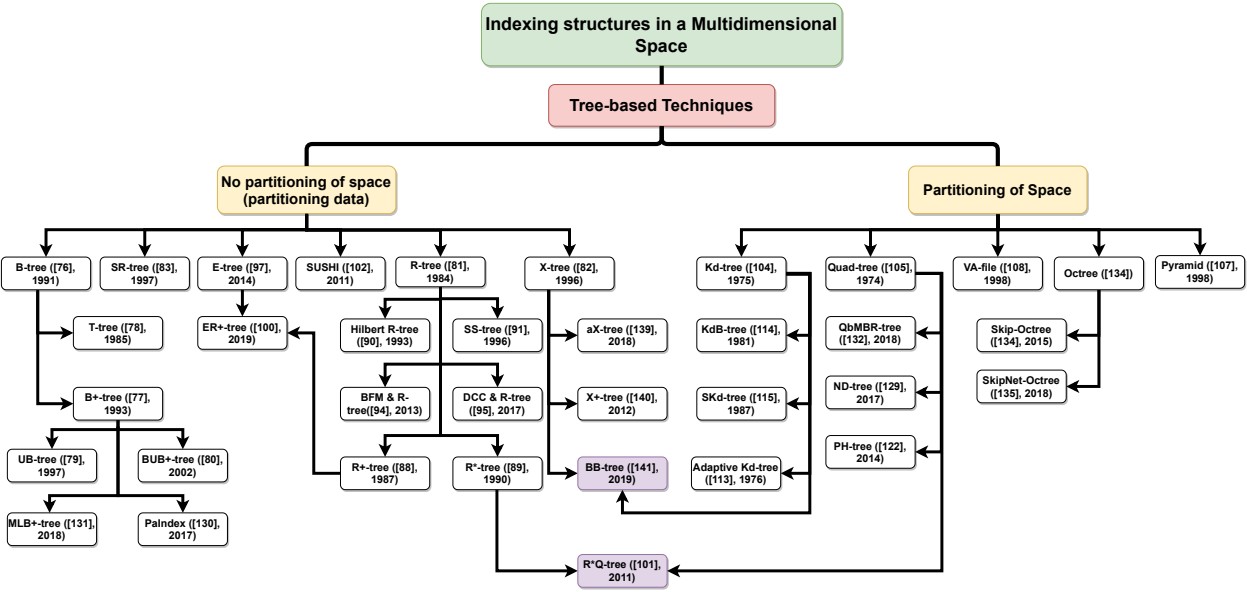

**Figure 8.** Taxonomy of tree-based indexing techniques.

## 7. A Comparative Analysis of Metric Access Methods

This section presents an overview of the advantages and disadvantages of metric access methods. Based on both approaches, partitioning and non-partitioning, a short taxonomy can be introduced [222,267,268] divided in two categories.

The first category does not use space partitioning, therefore, the family of the M-tree [53] generates a balanced incremental index. However, it suffers from the problem of overlap. An optimized version was proposed in [269]; this approach is the slim tree, which is based on the reorganization of the index to reduce overlap. Its disadvantage is the need to reinsert objects, which is costly.

As for the second category, it is based on the partitioning of space and two sub-approaches are provided: the first uses ball partitioning, such as VP tree [2], MVP tree [263], etc.; the other approach uses hyper plane partitioning, such as GH tree [219], GNAT [246], etc. The VP tree is based on sharing using balls. The MVP tree is a generalization of the VP tree. The nodes of the MVP tree are divided into quantiles. The CD tree [237] is a type of index based on hyperplane partitioning. It has proven its effectiveness for a limited number of dimensions.

Recently, a new technique has emerged MM tree [270,271], that also utilizes the partitioning by balls. An extension of this technique has been extended: the onion tree [3]. The objective is to separate the last region to generate successive enlargements, however, the problem is not fully resolved.

For more details, recall Figure 9 which presents a taxonomy, Tables 11–17 which provide an in-depth analysis, summarize the advantages and disadvantages, and application areas of metric indexing techniques.

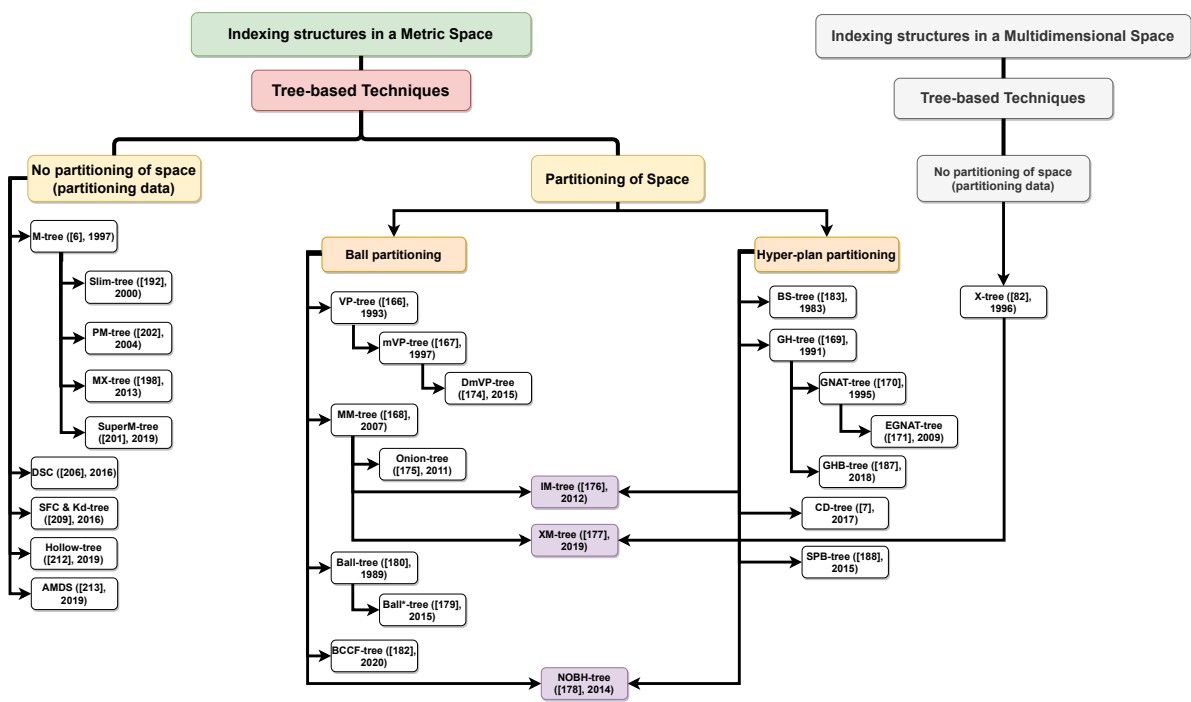

**Figure 9.** Taxonomy of tree-based indexing techniques in metric space.

## 8. Open Research Challenges

This review presented an overview of current indexing techniques and examined their advantages and disadvantages with respect to the large-scale perspectives of IOT data. In particular, the relevance of current indexing techniques for resolving deformities and responding to requirements was studied in detail. However, current indexing techniques still face some challenges. Therefore, in addition to the issues discussed above, several open problems are summarized in the following. At the end of the section, we summarize the most important future research directions for such challenges in Table 18.

**Table 18.** Summaries of open challenges and future directions.

| | Open Research Challenges | Future Research Directions |
|---|---|---|
| IoT Data Aggregation for 5G Data Indexing | • Reduced bandwidth<br>• Increased energy consumption<br>• Network congestion<br>• Network saturation | • Energy-balanced solution for cluster-based solution<br>• Development of future/5G networks<br>• Data networking solution based on emerging technologies (NFV, SDN, etc.) |
| Blockchain Data Indexing | • Centralized data storage<br>• Degradation of memory usage and block validation in IoT networks | • Towards user-friendly Blockchain Data Indexing |
| Security and Privacy for 5G Data Indexing | • No effective and confidential indexing of 5G data | • Design an indexation protocol for achieving privacy-preserving priority classification on 5G Data<br>• Enhance trust management for 5G networks via data indexing<br>• Secure the indexing approaches in 5G data indexing |
| Distributed Indexing for large-scale data | • The need for robustness, reliability, scalability, transferability and self-adaptation<br>• Reduce network bandwidth usage, overall cost and efficiency | • Distribute and balance system load across emerging IoT paradigms<br>• Towards multi-level indexing |
| IoT Data Representation in the Edge computing | • The different representations of IoT data and their damage in the processing and analysis of indexing structures | • Standard or unified architecture to provide connectivity to IoT devices, especially in the case of large-scale indexing |
| Indexing Software Processes | • Secure data during transmission | • Store encrypted data<br>• Analytical queries on encrypted Spatio-temporal data |

### 8.1. IoT Data Collection and Aggregation for 5G Data Indexing

Good quality data may increase the efficiency of data indexing. Therefore, data aggregation and corresponding incentives to increase data quality should be established. In the literature, several studies concentrate on data aggregation. In [272], Chen et al. introduce the first distributed aggregation technique for duty-cycle wireless sensor networks, and Zhuo et al. [273] describe a tripartite architecture for Mobile Crowdsensing (MCS) with fast data aggregation. Haiming et al. in [274] introduce a novel MCS system architecture that integrates a data aggregation and perturbation mechanism, and in [275], the authors suggest a payment mechanism that dramatically enhances the data quality in the MCS system. However, data aggregation issues, on the other hand, get more difficult as the number of data sources grows, necessitating more storage and computing capacity [276]. In addition, the expansion of the amount of digital data can result in reduced bandwidth, increased power consumption, and/or congestion imposed on the network, ultimately leading to network saturation due to a large amount of data being transmitted simultaneously over the network [49,277].

There are some new approaches such as cluster-based data aggregation algorithm for WSNs, but it's still ineffective because of their weak points like the problem of unbalanced energy dissipation [278]. With the development of future/5G networks and other emerging technologies such as Network Function Virtualization (NFV), the question that could be asked is, are these emerging technologies capable of developing improved and more energy-efficient approaches and improving the efficiency of data aggregation on IoT networks?

### 8.2. Blockchain Data Indexing

As cited in [279], blockchain technology can be applied effectively in almost all areas of IoT. Copies of the blockchain network ledger must be synchronized between all IoT entities, which could seriously affect memory usage and the effects of block validation, especially with the use of IoT networks. Consequently, the indexing of data in the block chain that supports decentralized storage becomes a difficult issue. In addition, indexing management has a significant role to play in improving the capabilities and efficiency of block chains for IoT. Therefore, the indexing of block chain data should be as user-friendly as possible.

### 8.3. Security and Privacy for 5G Data Indexing

Several factors such as the hardware used, operating system settings, buffer sizes, page sizes, and datasets affect the performance of a particular data structure in indexing 5G data. Therefore, since the real identity of the data could potentially be disclosed in 5G data indexing, critical security questions can be identified as follows:

- How to achieve efficient and privacy-preserving 5G data indexing?
- How to design an indexation protocol for achieving privacy-preserving priority classification on 5G Data?
- How to enhance trust management for 5G networks via data indexing in the era of big data?
- How to secure the multidimensional approaches in 5G data indexing (e.g., Pyramid, VA-file, $k$D-tree, X-tree, SR-tree, R*-tree, and R-tree)?

### 8.4. Distributed Indexing for Large-Scale Data

With the rapid development of IoT sensors, the requirements for robustness, reliability, scalability, transferability, and self-adaptation are higher. With the new computing paradigms that have emerged in the IoT arena (Cloud, Fog, Edge, Mist computing), distributed indexing systems offer a promising solution to solve the problem of search and discovery in Big IoT data. The latter consists of distributing the system load over the different layers of the system (from the sensor to the data center) through the different emerging paradigms. For the latter solution, several issues must be taken into account to create an efficient indexing structure:

- Is it possible to implement the structure at multiple levels?
- How to partition the indexing system load between these levels?
- To reduce the use of network bandwidth, overall cost, and efficiency, how to select the partition and the steps to be performed?

### 8.5. IoT Data Representation in the Edge Computing

It is important to note that the representation of the data in the IoT is different and that it can be stored in different formats [280]. The characteristics of the data in the edge computing of data collected from various IoT resources could cause serious damage when processing and analyzing the different IoT data models and structures. Therefore, the principal question that could arise is how to propose a standard or unified architecture to provide connectivity to IoT devices, especially in the case of large-scale indexing.

### 8.6. Indexing Software Processes

With IoT, big data comes private data, such as user locations and business data. This data must be encrypted on IoT devices before being sent to servers for storage and indexing. Although the data can be secured during transmission, it is usually stored after decryption. This is because every system assumes that only non-malicious users access the system and obtain the IoT data. However, many systems are currently attacked in a way that their private data is accessible. To date, it is best to:

1. Store encrypted data without decryption to maintain security.
2. To perform queries on encrypted data.

No IoT data analytics system supports queries on encrypted data. We can use analytical queries on encrypted data using existing techniques [281]. Nevertheless, these techniques are not optimal for Spatio-temporal data. The spread of IoT devices and applications has caused the emergence of secure IoT data analysis systems, but they have not yet been studied. There are also research opportunities for analytical queries on encrypted Spatio-temporal data.

## 9. Summary

The literature discussed in Section 5 presents an overview of the development of the indexing techniques proposed over the two decades. These techniques have been varied according to the nature of the data, the space and the nature of the structure as shown in the proposed taxonomy presented in Figures 6–9. The Tables 2–6, 8, 10, 12, 14 and 16 show that most of the structures studied have strong and weak sides.

The Tables 2–6, show that data independent hash methods are suffering from the high cost of space and time, making them effective for small data with low-dimensions and inefficient for large data with high-dimensions. For satisfactory results with data independent methods, many hash tables or long hash codes are required, which also makes them less effective in practice. Concerning data dependent hash techniques, supervised and semi-supervised hash methods take into account labeling information for training purposes, making them more effective than unsupervised hash methods because of the advantage of explicit semantic information in the data. On the other hand, they are much slower in terms of time/effort due to the higher cost of the training process, unlike unsupervised hash methods, which do not require any labelled data. Our study shows that unsupervised, supervised and semi-supervised hashes need new solutions to solve the problem of optimization to learn hash functions and hash codes mainly when data dimensionality increases. Due to the advantage of the higher power of the representation of the characteristics of the deep network, the deep hash achieves better performance than other hash methods. However, the non-optimal minimization of the quantization error is a major drawback that needs to redress [282,283].

The Bitmap index, in general, is intended for the optimization of search and retrieval of data with low variant, low cardinality and small distinct values (Family, Human, True,

False, etc.) and not for complex data where no particular property related to the nature of the data can be exploited.

The advantages and disadvantages/challenges of data indexing techniques in multidimensional and metric space are summarized in Tables 8, 10, 12, 14 and 16, where all approaches are discussed in Section 5. During the analysis of these indexing techniques, many factors influenced their operational and performance and proved their feasibility in certain applications.

One of the most important of these factors is the degree of data overlap. When distances tend to be very close to each other, objects become almost indistinguishable and cannot be clearly partitioned, which leads to a lot of overlap. A high degree of overlap leads to insufficient research output in terms of time, resources and quality because of the complete analysis of all data. This is confirmed by several researches, among them [229]. A review of all the techniques, showing that the technique that addressed the problem of partition overlap suffered from the overhead problem which influences real-time applications and especially on big data and vice versa.

The second major factor is the hardware. Despite all the advantages offered by the new high-performance computing such as cloud computing, fog computing and other emergence computations, there is not much work in the literature which adapts them. Knowing that It can provide many research opportunities to improve and develop more efficient indexing techniques (hashing, bitmap or tree). Which can take full advantage of the Graphics Processing Unit (GPU), Tensor Processing Unit (TPU) and Central Processing Unit (CPU) [64]. However, this does not mean that improving the performance of indexing structures depends only on increasing storage space and computing power.

At the end of this analysis, it can be concluded that the universal indexing technique that can manage arbitrary datasets has not yet been found and the realisation of this one which satisfies all the constraints and requirements of Big Data indexing mentioned in Section 4 is quasi-impossible.

## 10. Conclusions

A comprehensive review of the literature was presented in this paper, focusing on the indexing of large IoT data. In addition, an overview of the high requirements for data indexing was presented. The literature on indexing techniques for IoT data was also reviewed, analyzed, compared and classified in depth.

The authors also presented a comparative study of multidimensional indexing methods and a comparative study of metric access methods. Thus, several challenging areas of research can serve as a basis for possible future research directions for the indexing of large IoT data. We hope that this survey will be useful for researchers interested in indexing large IoT data.

**Author Contributions:** Conceptualization, Z.K., A.-E.B., M.A.F., B.F. and H.S.; methodology, Z.K., A.-E.B., M.A.F., B.F. and H.S.; software, Z.K., A.-E.B., M.A.F., B.F. and H.S.; validation, Z.K., A.-E.B., M.A.F., B.F. and H.S.; formal analysis, Z.K., A.-E.B., M.A.F., B.F., and H.S.; investigation, Z.K., A.-E.B., M.A.F., B.F. and H.S.; resources, Z.K., A.-E.B., M.A.F., B.F. and H.S.; data curation, Z.K., A.-E.B., M.A.F., B.F. and H.S.; writing—original draft preparation, Z.K., A.-E.B., M.A.F., B.F. and H.S.; writing—review and editing, Z.K., A.-E.B., M.A.F., B.F., H.S., M.K., A.A. (Adeel Anjum) and A.A. (Alia Asheralieva); visualization, Z.K., A.-E.B., M.A.F. and B.F.; supervision, Z.K., M.A.F., B.F., H.S., M.K., A.A. (Adeel Anjum) and A.A. (Alia Asheralieva). All authors have read and agreed to the published version of the manuscript.

**Funding:** This research received no external funding.

**Institutional Review Board Statement:** Not applicable.

**Informed Consent Statement:** Not applicable.

**Data Availability Statement:** Not applicable.

**Conflicts of Interest:** All authors declare no conflict of interest.

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
