# Peer review of "A Survey on Big IoT Data Indexing: Potential Solutions, Recent Advancements, and Open Issues"

_futureinternet, doi:10.3390/fi14010019_

Round 1
Reviewer 1 Report
General comments
This is a survey of the literature indexing systems for IoT data to consolidate the body of knowledge. It is distinguished from the other recent reviews cited. While the paper is substantial and well structured. I would suggest to add some discussion of the limitations of the study arising from its scope. In particular aspects of privacy and trustworthiness should be identified as largely beyond the scope of this survey.
Specific comments
- line 19-20 : focusses on volume, variety and velocity, but what about the value and veracity dimensions of big data? Given a variety of data sources, some categorization at least of the value / veracity dimensions should be useful.
- lines 19-20 systems that collect and administer data relevant to humans are likely to have issues with the privacy management of the data. GDPR and similar regulatory requirements apply in many jurisdictions. Even in the absence of those regulations, the data collected may be encumbered by other regulations or legal duties. Privacy may require administrative controls to modify or erase data. Some data may require anonymization before use. For more on Privacy metrics & Design
- Wagner, I., & Eckhoff, D. (2018). Technical privacy metrics: a systematic survey. ACM Computing Surveys (CSUR), 51(3), 1-38.
- Bu, F., Wang, N., Jiang, B., & Liang, H. (2020). “Privacy by Design” implementation: Information system engineers’ perspective. International Journal of Information Management, 53, 102124.
- lines 151/158-9 Figure 4 is referenced, but it is not clear what the reader is to take away from this. Is this figure asserting that everything big data is a subset of IoT? Is this figure asserting that IoT data needs additional metadata dimensions compared to other big data? Figure 4 also needs to be aligned with lines 19-20 which only mentions 3Vs
- lines 920-928 - blockchain indexing would be significantly affected by the structure of the blockchain e.g. Directed Acyclic Graphs in blockchains like IOTA tangle would be expected to have significantly different characteristics to more traditional blockchains like Ethereum.
- lines 929-940 - Sect 8.3security and privacy challenges are not restricted to 5G. See e.g.
- Wright, S. A. (2019, December). Privacy in iot blockchains: with big data comes big responsibility. In 2019 IEEE International Conference on Big Data (Big Data) (pp. 5282-5291). IEEE.
- Yu, B., Wright, J., Nepal, S., Zhu, L., Liu, J., & Ranjan, R. (2018). Iotchain: Establishing trust in the internet of things ecosystem using blockchain. IEEE Cloud Computing, 5(4), 12-23.
- lines 900-905 - Sect 8 - it is not clear what a challenge is. Are these personal opinions?, related to some performance objectives? In any way a complete list or systematically derived ? How Is it related to the constraints and requirements of big data indexing in Sect 4?
- lines 900-905 - Sect 8 - the challenges for the indexing software processes should also include consideration of the security/ trustworthiness/ confidentiality requirements associated with performing such indexing. See e.g.
- Bursell, M. (2021). Trust in Computer Systems and the Cloud. John Wiley & Sons.
Author Response
Please attached file.

Reviewer 2 Report
This study presents a systematic and detailed review, oriented towards indexing structures, of various data construction and extraction algorithms that are well-used in an IoT environment. However, there are several problems in the paper, which make the paper cannot be fully appreciated. Improvement is expected if the authors can take the following points into account.
- Big IOT Data is sourced from so many end devices such as Personal Computers (PC), smart phones, Global Positioning System (GPS) devices, sensors, and Radio Frequency Identification (RFID) devices, monitoring devices, etc. Also, online applications such as social networks and applications that involve video streaming are great sources that generate Big IOT Data. Big IOT Data collection is closely related to data indexing and should be introduced in the paper. [1] [2] are two important iot data collection surveys and should be quoted in the paper.
- Crowdsourcing is playing a more and more important role in Big IOT Data collection nowadays, the author should also introduce it in the paper. Crowdsourcing is used in [3] [4] for IOT data collection and should be quoted in the paper.
- Good quality data can improve the efficiency of data indexing. So relevant data aggregation and incentives which aim to improve the data quality also need to be introduced. [5] proposes the first distributed aggregation algorithm for duty-cycle wireless sensor networks, [6] introduces a three party architecture for MCS with efficient data aggregation. [7] introduces a novel MCS system framework that integrates a data aggregation, and a data perturbation mechanism, [8] proposes a payment mechanism which greatly improves the data quality in MCS system.
- Recent work on data indexing should also be taken into account, [9] [10] should be quoted in the paper.
In addition, some typos and grammatical mistake in this paper are listed as follows. The authors may check them out to improve the presentation.
- In page 4, line 46, “covering various definitions of IoT, core technologies, architecture and different applications” should be “covering various definitions of IoT, core technologies, architecture, and different applications”
- In page 4, line 48, “The study of [21] reviews the state of the art” should be “The study of [21] reviews the state of art”
- In page 8, line 170, “The constrains in Fig. 5” should be “The constraints in Fig. 5”
- In page 8, line 174, “ or differences elements” should be “or different elements”
- In page 8, line 182, “They focus of finding objects” should be “They focus on finding objects”
- In page 13, line 322, “such as: Convolutional Neural Networks” should be “such as Convolutional Neural Networks”
- In page 17, line 371, “R-tree is an hierarchical data structures” should be “R-tree is a hierarchical data structures”
- In page 24, line 589, “with less bits” should be “with fewer bits”
Overall, although this is a solid piece of work, a major revision is needed to improve the quality of this paper.
[1] N. C. Luong, D. T. Hoang, P. Wang, D. Niyato, D. I. Kim and Z. Han, "Data Collection and Wireless Communication in Internet of Things (IoT) Using Economic Analysis and Pricing Models: A Survey," in IEEE Communications Surveys & Tutorials, vol. 18, no. 4, pp. 2546-2590, Fourthquarter 2016, doi: 10.1109/COMST.2016.2582841.
[2] A. P. Plageras, K. E. Psannis, C. Stergiou, H. Wang, and B. B. Gupta,“Efficient iot-based sensor big data collection–processing and analysis insmart buildings,”Future Generation Computer Systems, vol. 82, pp. 349–357, 201.
[3] Y. Lu, A. Misra, w. Sun and H. Wu, "Smartphone Sensing Meets Transport Data: A Collaborative Framework for Transportation Service Analytics," in IEEE Transactions on Mobile Computing, vol. 17, no. 4, pp. 945-960, 1 April 2018, doi: 10.1109/TMC.2017.2743176.
[4] D. Y. Huang, N. Apthorpe, F. Li, G. Acar, and N. Feamster, “Iotinspector: Crowdsourcing labeled network traffic from smart home devicesat scale,”Proc. ACM Interact. Mob. Wearable Ubiquitous Technol., vol. 4,no. 2, jun 2020.
[5] Q. Chen, H. Gao, S. Cheng, J. Li and Z. Cai, "Distributed non-structure based data aggregation for duty-cycle wireless sensor networks," IEEE INFOCOM 2017 - IEEE Conference on Computer Communications, 2017, pp. 1-9, doi: 10.1109/INFOCOM.2017.8056960.
[6] G. Zhuo, Q. Jia, L. Guo, M. Li and P. Li, "Privacy-preserving verifiable data aggregation and analysis for cloud-assisted mobile crowdsourcing," IEEE INFOCOM 2016 - The 35th Annual IEEE International Conference on Computer Communications, 2016, pp. 1-9, doi: 10.1109/INFOCOM.2016.7524547.
[7] Haiming Jin, Lu Su, Houping Xiao, Klara Nahrstedt, "Incentive Mechanism for Privacy-Aware Data Aggregation in Mobile Crowd Sensing Systems", IEEE/ACM Transactions on Networking (TON), Vol. 26, No. 5, Pages 2019-2032, August 2018.
[8] Haiming Jin, Baoxiang He, Lu Su, Klara Narstedt, Xinbing Wang, "Data-Driven Pricing for Sensing Effort Elicitation in Mobile Crowd Sensing Systems", IEEE/ACM Transactions on Networking (TON), Vol. 27, Issue 6, Pages 2208-2221, December 2019.
[9] J. Xie, C. Qian, D. Guo, M. Wang, S. Shi and H. Chen, "Efficient Indexing Mechanism for Unstructured Data Sharing Systems in Edge Computing," IEEE INFOCOM 2019 - IEEE Conference on Computer Communications, 2019, pp. 820-828, doi: 10.1109/INFOCOM.2019.8737617.
[10] C. Wang, M. Xie, S. S. Bhowmick, B. Choi, X. Xiao and S. Zhou, "An Indexing Framework for Efficient Visual Exploratory Subgraph Search in Graph Databases," 2019 IEEE 35th International Conference on Data Engineering (ICDE), 2019, pp. 1666-1669, doi: 10.1109/ICDE.2019.00168.
Reviewer 3 Report
In this manuscript, the authors propose a survey on Big IoT Data indexing. In particular, the survey features several processes of thought in which the importance of Big IoT Data and its indexing is presented, first, and then a comprehensive study on indexing techniques, along with their pros and cons is presented.
The survey is clearly well organized. The structure proposed by the authors is solid and the contribution is well presented. The authors covered in detail a large number of works related to the context and performed a thorough comparison of them. The contribution is solid. However, in my opinion, the authors could improve the quality of the manuscript by addressing few concerns which are given below:
- The open research challenges are a useful addition to the survey; however, I suggest the authors to enclose them also in a table considering the most important features and research directions for such challenges. Such improvement could help the reader to quickly grasp what the challenges are.
- Section 7 might also take advantage of a more schematic way discussing it. Features, pros and cons of the metric access methods could be highlighted in a better way.
- A recent research context which can be addressed by Big IoT data indexing techniques is the so called Multiple IoT (or MIoT in short). The authors could refer to it both as a theoretical sandbox for the design of such techniques and as a real application example. The authors could consider to cite few pointers about it, such as https://doi.org/10.1016/j.pmcj.2020.101223.
Round 2
Reviewer 2 Report
The authors have addressed my comments to the previous version. I do not have further comments and recommend that this paper be accepted.
Reviewer 3 Report
The authors correctly addressed my concerns. In my opinion the manuscript now has a publication-acceptable quality.